ecology, evolution, palaeontology

carbon isotopes, diet, generalist, serial samples, specialist

**Author for correspondence:**
Larisa R. G. DeSantis
e-mail: larisa.desantis@vanderbilt.edu

# Global long-term stability of individual dietary specialization in herbivorous mammals

Larisa R. G. DeSantis[1,2], Melissa I. Pardi[2,3], Andrew Du[3], Michael A. Greshko[1], Lindsey T. Yann[2,4], Richard C. Hulbert Jr[5] and Julien Louys[6]

[1]Department of Biological Sciences, Vanderbilt University, 1210 BSB/MRBIII 465 21st Avenue S., Nashville, TN 37232, USA
[2]Department of Earth and Environmental Sciences, Vanderbilt University, 7th floor, Science and Engineering Building, 5726 Stevenson Center, Nashville, TN 37240, USA
[3]Research and Collections Center, Illinois State Museum, 1011 E. Ash St., Springfield, IL 62703, USA
[4]Waco Mammoth National Monument, 6220 Steinbeck Bend Drive, Waco, TX 76708, USA
[5]Florida Museum of Natural History, University of Florida, Gainesville, FL 32611, USA
[6]Australian Research Centre for Human Evolution, Griffith University, Brisbane, Australia

LRGD, 0000-0003-1159-9154; MIP, 0000-0001-9766-7511; AD, 0000-0002-7381-5089; LTY, 0000-0002-5040-9264; JL, 0000-0001-7539-0689

Dietary variation within species has important ecological and evolutionary implications. While theoreticians have debated the consequences of trait variance (including dietary specialization), empirical studies have yet to examine intraspecific dietary variability across the globe and through time. Here, we use new and published serial sampled $\delta^{13}C_{enamel}$ values of herbivorous mammals from the Miocene to the present (318 individuals summarized, 4134 samples) to examine how dietary strategy (i.e. browser, mixed-feeder, grazer) affects individual isotopic variation. We find that almost all herbivores, regardless of dietary strategy, are composed of individual specialists. For example, *Cormohipparion emsliei* (Equidae) from the Pliocene of Florida (approx. 5 Ma) exhibits a $\delta^{13}C_{enamel}$ range of 13.4‰, but all individuals sampled have $\delta^{13}C_{enamel}$ ranges of less than or equal to 2‰ (mean = 1.1‰). Most notably, this pattern holds globally and through time, with almost all herbivorous mammal individuals exhibiting narrow $\delta^{13}C_{enamel}$ ranges (less than or equal to 3‰), demonstrating that individuals are specialized and less representative of their overall species' dietary breadth. Individual specialization probably reduces intraspecific competition, increases carrying capacities, and may have stabilizing effects on species and communities over time. Individual specialization among species with both narrow and broad dietary niches is common over space and time—a phenomenon not previously well recognized or documented empirically.

## 1. Introduction

Food selection is a critical component of life at the individual, population and species level, and is one of the crucial aspects that defines the ecological niche of an organism. Diet influences an animal's habitat choice, landscape use/migration, how animals physically move in their environment (i.e. their biomechanics as pertains to food acquisition) and even reproduction [1–4]. The dietary niche of an animal is often then inferred as broad (i.e. a generalist) or narrow (i.e. a specialist) based on the breadth of food items consumed. Humans are no exception, with the genus *Homo* recently credited as occupying a previously empty niche—that of a 'generalist specialist', defined as the ability to both generalize (as a species) and specialize (as individuals) in one's

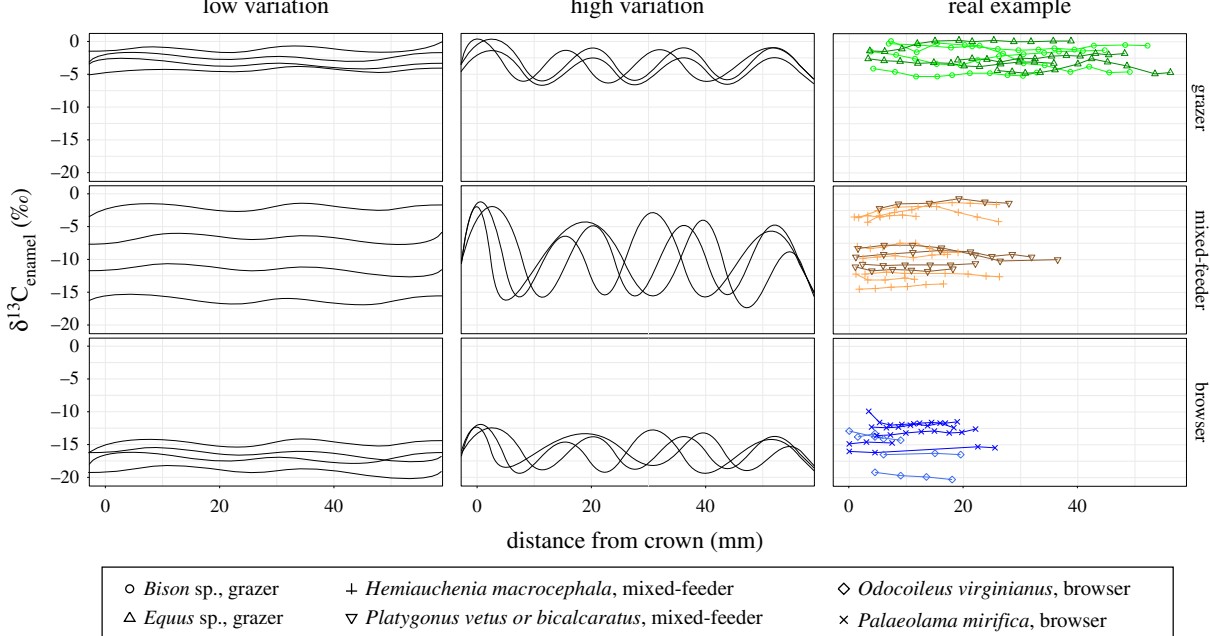

**Figure 1.** Stable carbon isotopic breadth of grazing, mixed-feeding and browsing species. Hypothetical examples denote expected individual variability if individuals are specialists (left) or more generalized (middle). Empirical data (right, examples of serial samples from electronic supplementary material, dataset) indicate that the average $\delta^{13}C_{enamel}$ range is 1.4‰ (1.1 s.d., $n = 21$; electronic supplementary material, tables S1 and S2); grazing (green), mix-feeding (orange), browsing (blue). (Online version in colour.)

environment [5]. While *Homo sapiens* are highly generalized in their dietary behaviour and individually maintain specialized diets through time, the 'generalist specialist' niche may be far more widespread than our species.

The niche variation hypothesis (NVH) suggests that populations of species with broader niches are also more variable in a particular trait (e.g. morphological, physiological and behavioural) than more specialized populations [6]. Increased niche breadth by a species could result from either individuals expanding their use of available resources or individual specialists maintaining narrow diets but collectively resulting in a broad species niche [7,8]. While empirical tests typically fail to support the NVH, the absence of confirmation has largely been attributed to the use of morphological traits (e.g. beak size or shape) as proxies of dietary breadth, in contrast with more direct measures of diet [8]. When diet is examined directly or inferred via proxies such as stable isotopes, generalist populations/species are often found to be composed of individual specialists—though only a handful of species have been examined [8–12]. The reason for this phenomenon is not entirely clear, but for carnivores it is likely to be related to tradeoffs between specializing on one versus many prey species, with selection acting against individual generalists [7]. For example, pursuing voles, rabbits and deer may make an individual predator less effective at capturing any one of these prey items, resulting in a 'jack of all trades and master of none' outcome. While this phenomenon has been relatively well studied in carnivores [9–13], herbivorous generalists have received less attention.

Food selection in herbivores results from the complex interplay between vegetation characteristics and herbivore anatomy, physiology and behaviour, a series of ecological and evolutionary interactions that occur at the species, population and individual levels [14]. Herbivore food selection is often a balancing act between the quality of digestible dry matter (energy and macronutrients) in each bite [15] and

chemical defensives such as phenolics, terpenes and toxins [16,17]. Individuals experience tradeoffs related to search time, nutrient content and fitness (e.g. optimal foraging theory [18–21]). Plant selection is further dictated by herbivore body size, through characteristics such as bite size and length of the digestive tract [17,22]. While foraging behaviour is relatively well studied in modern systems, few empirical studies have examined dietary variability in extant or extinct herbivores, or addressed how food selection at the individual level scales up to the species level. That is, are generalist species composed of similarly generalized individuals or specialized individuals subdividing the species' dietary niche axis? How general is either pattern over time?

Here, we use stable carbon isotope data from serial samples of mammal tooth enamel ($\delta^{13}C_{enamel}$) collected from herbivores that can be categorized into the dietary groups of grazer, browser and mixed-feeder (i.e. eating both grass and browse). We gathered data from the literature dating from the Miocene to the present (298 individuals, 4013 samples) and acquired new samples of mixed-feeders (20 individuals, 121 samples). Enamel $\delta^{13}C$ values provide valuable insights into herbivore ecology, including a record of the proportions of $C_3$ and $C_4$ vegetation consumed, indicative of trees/shrubs and warm-season grasses, respectively—when occurring at lower latitude sites (approx. 37° and below [23–25]). Because teeth grow incrementally, serial samples of enamel collected perpendicular to a tooth's growth axis record dietary variability over the course of the tooth's growth which can range from a few months to over 2 years in high-crowned teeth [26,27]. As a result, our new dataset allows us to answer the following questions. (i) Are herbivorous generalists (i.e. mixed-feeders) composed of individual generalists (defined as having high isotopic variation) or individual specialists (defined as having low isotopic variation) (figure 1)? And (ii) how do isotope values of individuals vary within a species's overall dietary strategy

**Table 1.** Summary statistics of dietary ranges for individuals and subsequent comparisons between groups using Kruskal–Wallis and Dunn's tests. s.d., standard deviation; *n*, number of individuals. Italicized *p*-values are significant, <0.05.

| dietary category | mean | s.d. | *n* | comparison | *p*-value |
|---|---|---|---|---|---|
| *global dataset* | *range* | | | | |
| browser | 1.1 | 0.8 | 48 | browser versus grazer | *<0.001* |
| mixed-feeder | 1.4 | 0.9 | 33 | mixed-feeder versus browser | 0.0629 |
| grazer | 1.8 | 1.5 | 231 | grazer versus mixed-feeder | 0.3785 |
| *below 37° latitude* | *range* | | | | |
| browser | 1 | 0.7 | 19 | browser versus grazer | *<0.00001* |
| mixed-feeder | 1.5 | 1.0 | 32 | mixed-feeder versus browser | 0.0815 |
| grazer | 2.4 | 1.7 | 111 | grazer versus mixed-feeder | *<0.01* |

(i.e. browser, mixed-feeder, grazer)? These answers provide a long-term view of herbivore-vegetation interactions critical for understanding intra- and interspecific competition and their ecological and evolutionary consequences [23,28]. Understanding how herbivores choose and consume vegetation at the individual, population, and species levels is also fundamental for effective environmental conservation and management [29].

## 2. Results

Descriptive statistics and raw data of $\delta^{13}C_{enamel}$ values (4134 serial samples from 318 individuals) are noted in table 1, and electronic supplementary material, tables S1–S11, and summarized in figures 1–4 and electronic supplementary material, figures S1–S3.

Mammalian herbivore $\delta^{13}C_{enamel}$ values spanned 23‰ (ranging from −20.3‰ to 2.7‰) across all individuals, which includes specimens from grasslands to rainforests across the globe (e.g. Afghanistan, China, Ethiopia, Ireland, Panama, South Africa and USA; see dataset in the electronic supplementary material for a full list of countries included). The $\delta^{13}C_{enamel}$ range within a given species can be upwards of 8‰ in grazers (figure 3), or approximately 35% of the total range represented in this study. By contrast, the range within each individual specimen is low (typically less than 3‰; figure 3). For example, the total species range of the mixed-feeder *Cormohipparion emsliei* from the early Pliocene (Florida, USA [30,31]) is 13.4‰, while individuals have an average $\delta^{13}C_{enamel}$ range of 1.1‰ (standard deviation, s.d. 0.5‰, $n = 9$; figure 2), and all individuals vary by ≤ 2‰. An additional mixed-feeder targeted for sample collection (*Hemiauchenia macrocephala*) also demonstrates a broad $\delta^{13}C_{enamel}$ range of 13.2‰ from two sites in Florida (Inglis 1A and Leisey 1A [32]), with an average individual $\delta^{13}C_{enamel}$ range of 1.3‰ (s.d. of 0.5‰, $n = 8$), and all individuals exhibit ≤ 2.4‰ variability (figure 1). All individual generalist herbivores (i.e. mixed-feeders) have $\delta^{13}C_{enamel}$ ranges ≤ 4.0‰ with average individual $\delta^{13}C_{enamel}$ ranges of less than 2‰.

Nearly all herbivorous mammalian individuals are specialized in their diets (figure 3). Specifically, the majority of individuals sampled with at least three serial samples (89%, 283 of 318 individuals) or five or more serial samples (87.9%, 246 of 280 individuals) exhibit a $\delta^{13}C_{enamel}$ range of

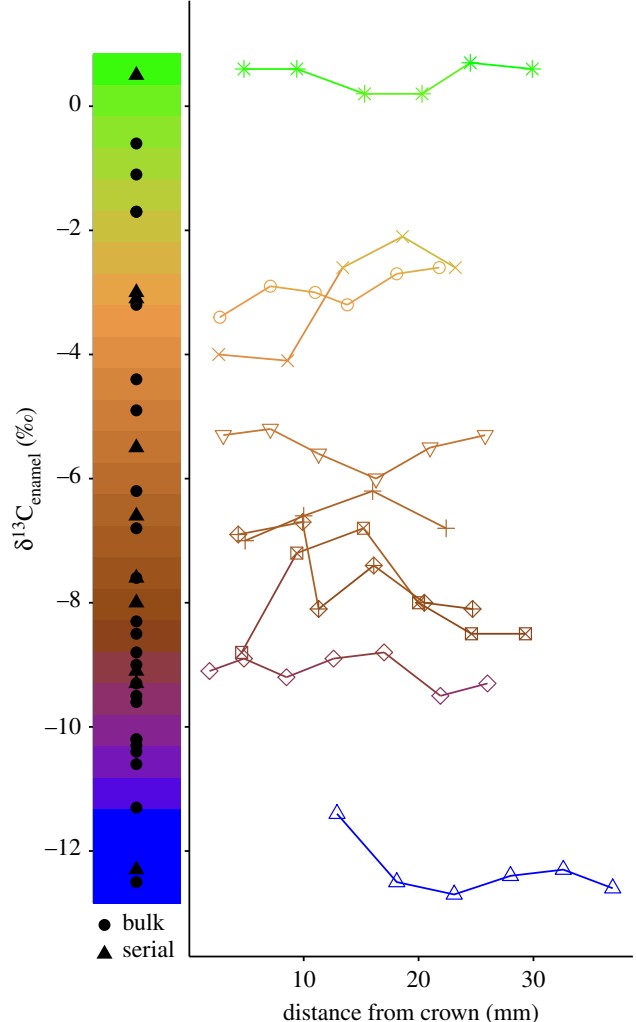

**Figure 2.** Carbon stable isotope values from individual specimens of *Cormohipparion emsliei* from a 5-million-year-old fossil assemblage (Bone Valley, Florida, USA). Bulk and mean serial values (left) along with raw serial samples (right) exemplify broad dietary breadth as a species (range = 13.4‰), while individuals are highly specialized (all individuals sampled have less than or equal to 2‰ range of $\delta^{13}C_{enamel}$, with an average $\delta^{13}C_{enamel}$ range of 1.1‰; electronic supplementary material, tables S2 and S3). Browsers (blue, bottom), grazers (green, top) and all remaining mixed-feeders are indicated with other colours (orange-purple, middle). (Online version in colour.)

less than or equal to 3‰ (figure 3). As body size is known to influence diet, including an animal's ability to eat lower quality foliage like grass [33], relationships between

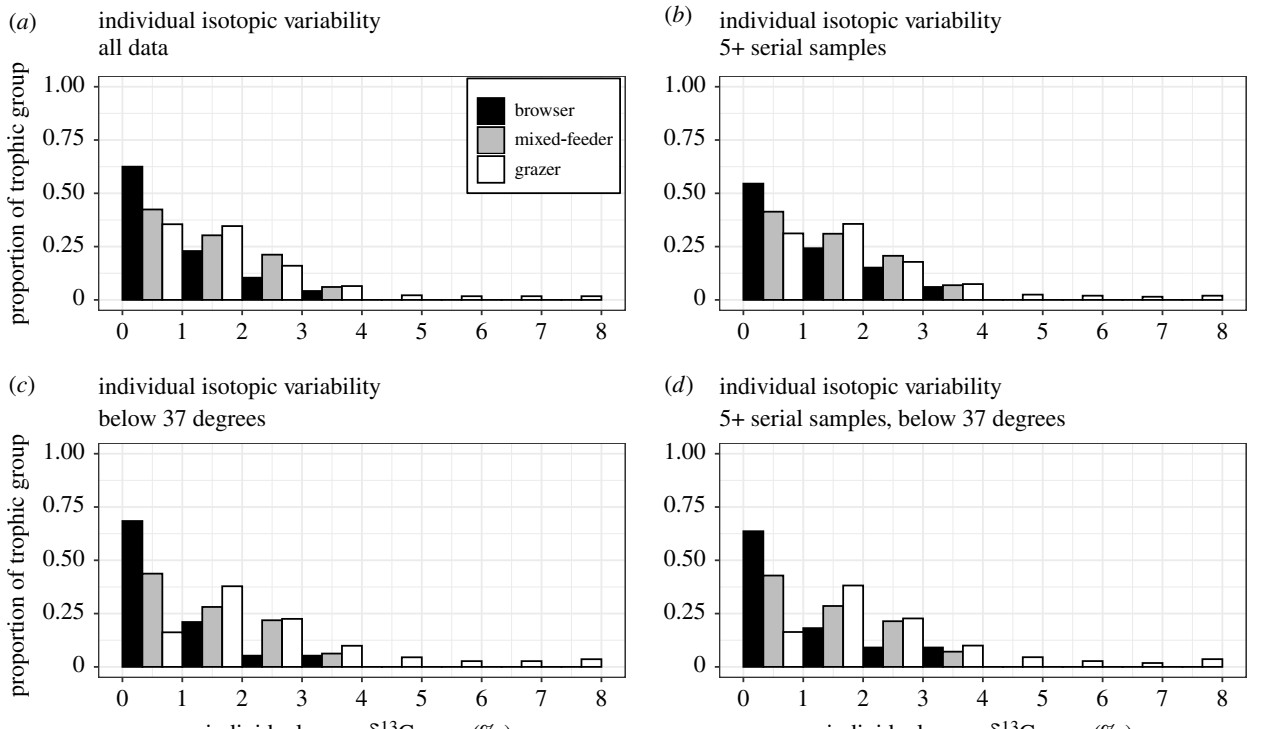

**Figure 3.** The proportion of individuals within serial sample range bins per dietary group. Most serially sampled individuals (82–89%), regardless of dietary category, have $\delta^{13}C_{enamel}$ values that range less than or equal to 3‰ when calculated using the full range of statistical methods. Only a small proportion of individuals (3.8–6%) range in $\delta^{13}C_{enamel}$ values by more than 5‰, all of which are grazers. The right-skewed pattern is present whether we use (a) the entire dataset, (b) individuals with 5 or more serial samples, or when we consider (c) all samples below 37° latitude, and (d) individuals that have at least 5 serial samples that are also restricted to below 37° latitude.

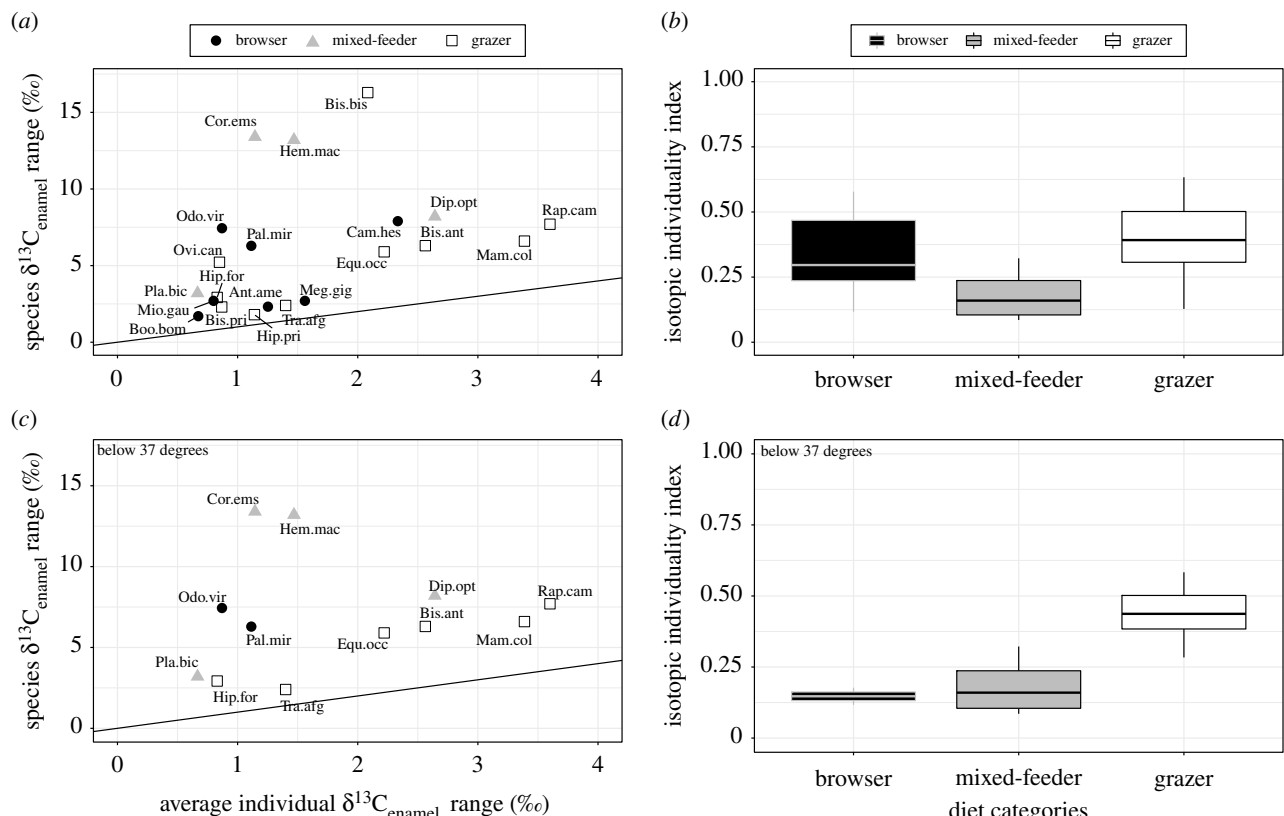

**Figure 4.** Isotopic individuality index (III) per dietary category. (a) The overall species dietary range plotted against the average individual range for a species (one-to-one line given, species codes are defined in the electronic supplementary material, table S9) and (b) the distribution of III within dietary categories indicate a high degree of individual specialization (low index). Values closer to one are species in which the individuals are each more representative of the species as a whole, while smaller values indicate that individuals are specialized and less representative of their overall species dietary breadth. (c) Below 37° latitude grazers plot near the one-to-one line more often than other groups; (d) the III for grazing species is higher than mixed-feeders below 37° latitude, and browsers at these lower latitudes are uncommon in this study.

individual breadth in $\delta^{13}C_{enamel}$ and body size were examined. We tested for significant differences in the total range of $\delta^{13}C_{enamel}$ values of small (mass less than 100 kg), medium (between 100 kg and 350 kg) and large (mass greater than or equal to 350 kg) species, as well as the average range exhibited by individuals across these size categories (see Material and methods). We also reran our analyses on only specimens found below 37° latitude, where the isotopic distinction between grasses (primarily $C_4$) and trees/shrubs (primarily $C_3$) is clearest [23–25]. There are no significant relationships between average individual ranges or of average taxon $\delta^{13}C_{enamel}$ ranges across body size categories, either globally or at low latitudes (electronic supplementary material, table S8), contrary to expectations [34]. However, when data were corrected with a sliding window (due to the possibility that more serial samples could produce larger $\delta^{13}C_{enamel}$ ranges; see Material and methods), the average individual range of large taxa is significantly greater than that of medium-sized taxa ($p < 0.05$), but not of small taxa ($p > 0.42$; electronic supplementary material, table S8).

The isotopic individuality index (III) for a given species, as defined here, is the ratio of the average individual $\delta^{13}C_{enamel}$ range to the total $\delta^{13}C_{enamel}$ range of a species. This proportion is low (near 0) when individuals are specialized within isotopically generalized species; the III approaches 1 when an individual's isotopic breadth approaches the overall species breadth (figure 4). The average IIIs calculated for mixed-feeding species ($0.18 \pm 0.11$, $n = 4$), browsers ($0.35 \pm 0.17$, $n = 7$) and grazers ($0.39 \pm 0.17$, $n = 10$) are all closer to 0 than 1, with only grazers having significantly higher III values than mixed-feeders when including the global dataset ($p < 0.05$) (figure 4; electronic supplementary material, tables S2 and S7). Below 37° latitude, the III of grazers is significantly higher than both mixed-feeders ($p < 0.05$) and browsers ($p < 0.05$). It should be noted that sampling standardization (i.e. the sliding window analysis) of the global dataset results in significantly lower III values in mixed-feeders as compared to browsers ($p < 0.01$; electronic supplementary material, tables S8 and S9). Variance partitioning analyses (where the proportion of $\delta^{13}C_{enamel}$ variance is partitioned across three nested scales: between species, between individuals and within individuals) [35] similarly found that, across dietary groups, individuals within species tend to be specialized (electronic supplementary material, table S1). Lower variance is consistently found within individuals as opposed to among individuals in a species, or across species: depending on the specific analysis used, the proportion of variance ranges from 0.34 to 0.64 between species, 0.32–0.61 between individuals within a species and 0.05–0.06 between serial samples within individuals (electronic supplementary material, table S1).

Across the global dataset, the average individual $\delta^{13}C_{enamel}$ range is highest among grazers, significantly higher than browsers (table 1). Below 37° latitude, the average individual $\delta^{13}C_{enamel}$ range remains highest among grazers and is significantly higher than both browsers and mixed-feeders (table 1). All dietary categories have mean $\delta^{13}C_{enamel}$ ranges of less than 3‰, grazers yielded a maximum individual range of 8‰, while the individual $\delta^{13}C_{enamel}$ ranges of browsers and mixed-feeders in the dataset never exceeded 3.3‰ and 3.9‰, respectively (figure 3). When we standardize the number of analysed serial samples per tooth using a five-sample sliding window approach (see Material and methods), we find similar results to the raw data analysis (electronic supplementary material, figure S1 and table S7). Most individual ranges (92–94%) are less than or equal to 2‰ when standardized, and nearly all (97–100%) are less than or equal to 3‰ across dietary groups (i.e. global average; electronic supplementary material, figure S1 and table S7).

## 3. Discussion

### (a) Effects of dietary behaviour on individual specialization

Herbivores that are classified as grazers vary their diet the most individually, more so than herbivores classified as browsers and mixed-feeders. Some of the largest individual $\delta^{13}C_{enamel}$ ranges are exhibited by grazers (e.g. bison, horses, mammoths and wombats can exhibit individual $\delta^{13}C_{enamel}$ ranges of 6.7‰, 6.7‰, 7.6‰ and 8.0‰, respectively; electronic supplementary material, figure S3); however, their mean is still fairly constrained ($\bar{x} = 1.8‰ \pm 1.5$ s.d.). As grazing morphologies permit (but do not exclusively prescribe) grass consumption [36], dietary variability of individuals is likely to be broader when a given taxon is capable of eating grass as well as browse. Browsing species are morphologically and nutritionally constrained to diets that exclude grasses [37] and exhibit the lowest $\delta^{13}C_{enamel}$ ranges (figure 3 and table 1), which could explain the broader $\delta^{13}C_{enamel}$ ranges of grazers as compared to browsers (approx. 1.6 to 2.4 times that of browsers; table 1).

The manner in which grass and browse resources are exploited by herbivores can affect $\delta^{13}C_{enamel}$ ranges among grazer and browser species. Specifically, mammalian herbivores broadly record $\delta^{13}C_{enamel}$ variability of plants on the landscape; depending on the size of their home range, this could introduce spatial variability in $\delta^{13}C_{enamel}$ that may be expected to increase with body size. While many large herbivores are classified as grazers, and body size of the individual also impacts the plant types, parts and volume that can be ingested [34,37], a range of body sizes are represented in each dietary category. Further, our results suggest that body size alone is not significantly influencing individual specialization. While small herbivores were analysed (less than 100 kg), the smallest mammals (less than 10 kg; e.g. most rodents) are not included in this analysis due to the size required for serial sampling of individual teeth. While more work is needed on small mammals, modern mark–recapture studies suggests that small mammals can also be individually specialized [38].

Fundamental physiological differences between grasses and browse, and their distribution across the landscape, may influence spatial heterogeneity and variation in $\delta^{13}C_{enamel}$ values. Grasses typically exhibit fewer secondary plant chemicals (such as tannins and alkaloids) that can change more seasonally than leaves and buds [17], contributing few tradeoffs for their consumption if herbivores can compensate for their typically lower nutritional content and abrasive nature (i.e. the presence of phytoliths which can wear down teeth [39]). $C_4$ grasses that are better adapted to aridity and use the PEP-CK sub-pathway can also have more negative values than grasses that use the classic NADP sub-pathway, though differences are small (approx.

1‰) [38]. From an ungulate perspective, the architectural arrangement of grass cannot be as easily differentiated via dietary selection as can browse—grasses are composed of leaves, sheaths and fruit that differ only at the very fine scale, they have a low growth form, and grow in continuous dispersion on the landscape [17], making it less likely that individual grazers (typically with broad muzzles) can more finely select grass forage. In contrast, browse consists of a heterogeneous mix of leaves, buds, and woody stems that are irregularly distributed over the plant [17]. Therefore, active food selection of browse resources can lead to narrower dietary niches in browsing and mixed-feeding taxa [37] but is unlikely to be the source of higher $\delta^{13}C_{enamel}$ variation in species consuming primarily grasses. The higher individuality indices of taxa classified as grazers as compared to other dietary groups likely stems from their ability to eat diverse food types in addition to grasses (with few consequences).

Mixed-feeders, despite eating the broadest mix of food types, have the lowest III values (significantly less than grazers; electronic supplementary material, table S2). We would expect lower III values for mixed-feeders if individual variation is comparable to grazers or browsers, due to the overall larger breadth represented by mixed-feeding species. However, individual specialization of mixed-feeders is far below that represented by the species and does not indicate examples of switching between the consumption of primarily grass at one interval to the consumption of primarily browse at another. Lastly, while III values of fossil taxa could be lower than extant taxa due to time averaging that contributes to species breadth calculations (i.e. the denominator), the vast majority of all specimens included (approx. 96%) are from fossil specimens and are relatively comparable to one another (electronic supplementary material, dataset SD1).

It is compelling that the vast majority of mammals exhibit narrow isotopic variability for comparable durations (months/seasons) of an individual's life (figure 3 and table 1). We can't discount the possibility that herbivores change their diet later in life or over the course of multiple years to decades; however, serial sampling of multiple teeth per individual (i.e. spanning years, though not decades) reveals the absence of significant dietary variability in the majority of specimens (though this has only been documented in a handful of studies [40–42]). Although stable isotopes in teeth are known to be dampened compared to the isotopic composition of diet or water consumed [26,43,44], high variability is still possible (electronic supplementary material, figure S3). That being said, serial samples represent an averaged diet over the period of time sampled and do not represent the full isotopic range of plants consumed by the individual during this period of time, especially if mixed-feeders are consistently consuming the same proportions of mixed vegetation. While the total $\delta^{13}C_{enamel}$ range from an individual tooth is only a minimum estimate of total plant isotopic variability, it does provide important insight into dietary variability of individuals and is a useful tool for comparing taxa with different dietary preferences as dampening is likely to affect herbivorous mammals to a similar degree. Mixed-feeders with broad isotopic ranges (e.g. *Hemiauchenia macrocephala*, *Platygonus vetus* and *Diprotodon optatum* [32,45]) individually exhibit only a fraction of the isotopic variability of a given taxon (figure 4). Most notably, very few individuals (1.2%) exhibit a broad isotopic range (i.e. individual $\delta^{13}C_{enamel}$

range greater than 7‰), demonstrating both the possibility of high $\delta^{13}C$ variability in an individual, but also the rarity of such occurrences.

## (b) Dietary specialists across space and time

Our results suggest that herbivorous mammals are primarily individual specialists regardless of dietary category, similar to observations noted for carnivorous taxa [9–13]. The individual variation we observe in herbivorous mammals across the globe and through time have important ecological and evolutionary implications [8,28,46,47]. Fundamentally, trait variation among individuals is the raw material natural selection acts upon [48]. Individual foraging behaviour can determine how other members of a foraging group behave, a group's choice of where to forage, and foraging methods and their successes [49–52], while individual temperament can affect predator–prey interactions [53]. An animal's temperament and the individual dietary choices it makes (including high degrees of individual dietary specialization) are often associated with ecological interactions that promote food web stability in diverse communities [53]. Specialization of individuals on disparate foods within a given location can reduce competition among conspecifics [54], while also potentially promoting predator–prey stability for high carrying capacities of prey [55]. Specifically, individual specialization may allow for a higher carrying capacity than otherwise possible due to the reduction in competition both among and between species (niche complementarity) [56]. Alternatively, trait heterogeneity may be a consequence of a release of intraspecific competition resulting in increased niche widths in populations that are decoupled from individual niche width [57].

Generalist herbivore species are mostly composed of individual specialists and not individual generalists (figure 1). If generalists were composed of individual generalists, one might expect fitness tradeoffs with the 'jack of all trades but master of none' strategy. Analogously, foraging behaviour among herbivores may have similar tradeoffs pertaining to an individual animal's ability to search for and digest disparate food types [58]. Generalist populations or species composed of individual specialists may have overall increased resilience against extinction if vegetation, patch size and conditions affecting foraging behaviour change dramatically over time. Notably, species with high functional heterogeneity of dietary behaviour and/or other functional traits have a higher probability of persisting as environmental conditions change—having a stabilizing effect that may result in increased species longevity [28,59]. These observations have important conservation implications, as species survival for threatened herbivores may hinge on managing populations and landscapes in ways that place dietary specialized individuals in different habitats in order to preserve species-level generalization—an urgent mandate, given the existential threats facing many mammalian herbivores around the world [60].

Herbivores select food in a hierarchical fashion, with individuals and populations making decisions at different spatial scales. At the individual level, this relates to the size of the bite and the plant part exploited, and extends to the landscape and regional scale for populations [17,37]. Our data extends this spatial perspective to include specimens from the Miocene to the present. Individual foraging decisions in

herbivores are largely consistent through the time of tooth mineralization but are not representative of the realized niche of the species (figure 4). This suggests that individual temperament and/or learned behaviour may have contributed to the way herbivores exploit plants throughout the geological record, and that ecological processes similar to those at play today (e.g. intraspecific competition, tradeoffs in resource exploitation) have fostered individual specialization within generalist species through millions of years, a pattern with profound evolutionary implications. Across time and space, generalist herbivore populations are not 'jacks of all trades and masters of none'; because each individual 'jack' specializes in a dietary 'trade,' populations can become 'jacks of all trades and masters of all'. Individual specialization is not limited to generalist species but instead occurs in the vast majority of extant and extinct mammals examined here, suggesting there is a selective benefit of individual specialization. Far from being a characteristic unique to *Homo sapiens*, the 'generalist specialist' niche can be viewed as a ubiquitous characteristic of generalists that may help promote the stability of complex and diverse communities and have the potential to contribute to species longevity.

# 4. Material and methods

## (a) Stable isotope analyses

Published data were obtained through a literature search of the topics of 'isotop* AND fossil' or 'isotop* AND tooth' through Web of Science. Publications with serial sample data ($\delta^{13}C_{enamel}$ from the carbonate portion of tooth enamel hydroxylapatite) from mammal teeth around the globe and across the Neogene were curated in a database (electronic supplementary material, dataset SD1, referred to as the ISOSERIAL 1.1 database). The identity of the sample (including the published taxonomy along with any revisions), measured $\delta^{13}C_{enamel}$ values, the distance each sample was taken along with the growth axis of the tooth, and the publication reference were recorded, when available. Additional metadata, including the museum collection ID and specimen number, the absolute or relative age of the specimen and location data (name, country, and state of the site the specimen originated from) were collected. Some publications only reported summary statistics from the isotope values for a given study (e.g. minimum, maximum, mean, standard deviation and range); these were recorded and indicated separately, along with their corresponding metadata (electronic supplementary material, dataset SD1). Original data, specific to this publication, are indicated as such (electronic supplementary material, dataset SD1, ISOSERIAL 1.1 database). Domesticated and zoo specimens, marine mammals and xenarthrans were excluded from data collection (the latter which lacks tooth enamel). A total of 3330 raw serial samples are included in the database with summary values recorded for 332 individuals (4134 serial samples), including 121 new stable isotope serial samples from 20 individual mixed-feeders (electronic supplementary material, tables S3–S5 and dataset SD1, ISOSERIAL 1.1 database). The focus of this dataset is medium- to large-sized herbivores (i.e. ungulates and proboscideans). The majority of all data included in this database are from herbivores that occur during or after the expansion of $C_4$ grasses in their region, with only approximately 7% of data occurring in low-latitude regions prior to the expansion of $C_4$ grasses, in the early–middle Miocene (based on [23]).

Of the new data collected for this publication, mixed-feeding taxa were targeted for serial sample analysis as they were rare in published literature and are of particular relevance to the research questions. All newly sampled specimens are noted in the electronic supplementary material, tables S3–S5 and dataset SD1, and are housed in publicly accessible collections in the Florida Museum of Natural History (Gainesville, FL, USA). Enamel powder (approx. 1–2 mg) was sampled perpendicular to the growth axis of the tooth with spacing indicated in the electronic supplementary material, dataset SD1 and table S4. Enamel powder was treated with 30% hydrogen peroxide, rinsed, reacted with 0.1 N acetic acid and rinsed again after 18 h (per [32]). The treated and dried enamel powder was analysed using a VG Prism stable isotope ratio mass spectrometer with an in-line ISOCARB automatic sampler in the Department of Geological Sciences at the University of Florida. The analytical precision is $\pm 0.1‰$, based on replicate analyses of samples and standards (NBS-19). Stable isotope data are reported in conventional delta ($\delta$) notation for carbon ($\delta^{13}C$), where $\delta^{13}C$ (parts per mil, $‰) = ((R_{sample}/R_{standard})-1) \times 1000$, $R = {}^{13}C/{}^{12}C$, and the standard is VPDB (Pee Dee Belemnite, Vienna Convention) [61]. All stable carbon isotopes are from the carbonate portion of tooth enamel hydroxylapatite.

## (b) Dietary classification

Each taxon was categorized as a browser, grazer or mixed-feeder according to literature consensus. For extant species, this was determined by the observed diet of wild-caught specimens (note, only wild-caught specimens were included in the database; hence, no domesticated species were included from modern or archeological sites). For extinct species, these determinations were made based on studies of analogous morphology with living species. Taxon names, diets and the references used to justify the assignments for extinct taxa are given in the electronic supplementary material, table S12.

## (c) Statistical analyses

The minimum, maximum, range, mean and standard deviation of the measured $\delta^{13}C_{enamel}$ serial samples were calculated for each individual (electronic supplementary material, dataset S1). When serial samples were from multiple teeth from the same individual, summary statistics were calculated from the combined serial samples. Individuals were analysed if they had at least three serial samples for a given tooth.

To explore whether diet type influences individual $\delta^{13}C_{enamel}$ breadth, individual range was categorized into 1‰ bins (i.e. $0 < x \leq 1‰$, $1 < x \leq 2‰$, etc.), and the proportion of individuals occupying each dietary type was calculated. To assess the influence of body size on individual breadth in $\delta^{13}C_{enamel}$, we tested for significant differences in the range of $\delta^{13}C_{enamel}$ for species of small (mass less than 100 kg), medium (100 kg $\leq$ mass less than 350 kg), and large (mass greater than or equal to 350 kg) body size as well as the average range of individuals across these size categories. These categories are based on the commonly used and well-established body size categories of herbivorous mammals, based in part on dietary differences and the prey sizes readily consumed by African predators [62,63]. Body size estimates were made from the published literature, or new estimates were made based on allometries (electronic supplementary material, table S13). The potential influence of body size was assessed for the global dataset, the global dataset corrected for serial sampling bias, the dataset restricted to low latitudes (less than 37°) and the low-latitude dataset corrected for serial sampling bias. Significant differences between average individual ranges according to dietary type was tested using Dunn's test of non-parametric pairwise multiple comparisons.

Grazers tend to have higher crowned teeth than browsers and mixed-feeders and therefore tend to have more serial samples per tooth. As sampling more serial samples might produce a larger range of $\delta^{13}C_{enamel}$ values, we performed a moving window

analysis for each tooth, where we iteratively calculated the $\delta^{13}C_{enamel}$ range for five consecutive samples, as denoted by each sample's distance from root along the tooth's growth axis.

An isotopic individuality index (III) was calculated for each taxon, as follows: average individual range of $\delta^{13}C_{enamel}$ (for a given taxon)/Total $\delta^{13}C_{enamel}$ range of the taxon. Average IIIs for dietary categories were calculated using only species-level data, unless it could be reasonably inferred that a genus was represented in the database by only one species. Average IIIs were calculated at two scales: (i) at the global scale and (ii) restricted to individuals collected below 37° latitude (where $C_4$ resources are primarily warm-season grasses and $C_3$ resources are primarily trees and shrubs, in contrast with $C_3$ ecosystems above 37° latitude, where grass and browse cannot be discerned from isotopes alone [24,25]). While we recognize that the latitudinal gradient in $C_3$ and $C_4$ grasses is likely to be a recent phenomenon, since approximately 5–7 Ma [23,64], there is no evidence from our analysis that any taxon is switching from $C_3$ to $C_4$ grass, seasonally or during the period of time during which their teeth mineralize (i.e. no teeth vary by approximately 14‰, the mean difference between $C_3$ and $C_4$ grasses [23]). Thus, the analysis of $\delta^{13}C_{enamel}$ values of individuals collected below 37° latitude is a reasonable proxy for inferring individual dietary variability.

We conducted variance partitioning analyses to quantify the proportion of $\delta^{13}C_{enamel}$ variance that can be found between species, between individuals and within individuals. We only analysed those teeth with at least five serial samples. Individuals identified to the genus level were lumped with congener individuals identified to the species level only when it could be reasonably inferred based on location and geological time-period that they belonged to the same species. Variance partitioning was done in two ways using R v. 4.0.2 [65]. (i) We fitted via restricted maximum likelihood an intercept-only,

nested ANOVA with $\delta^{13}C_{enamel}$ as the dependent variable and individuals nested within species as random effects. This was done using the lme() function in the 'nlme' R package [64], and the varcomp() function in the 'ape' package [66] was used to extract the variance components. (ii) We also partitioned variance (sum of squares) in $\delta^{13}C_{enamel}$ using two-level, nested ANOVAs for unbalanced data following [35] (their box 10.6, pp. 294–298).

Ethics. All new data were sampled from publicly accessible collections. All published data are from the literature.

Data accessibility. All primary data have been provided as electronic supplementary material [67].

Authors' contributions. L.R.G.D.: conceptualization, data curation, funding acquisition, project administration, resources, supervision, validation, visualization, writing—original draft, writing—review and editing; M.I.P.: conceptualization, data curation, formal analysis, funding acquisition, investigation, methodology, resources, validation, visualization, writing—review and editing; A.D.: formal analysis, methodology, validation, writing—review and editing; M.A.G.: conceptualization, data curation, methodology, writing—review and editing; L.T.Y.: data curation, writing—review and editing; R.C.H.: data curation, writing—review and editing; J.L.: investigation, writing—original draft, writing—review and editing.

All authors gave final approval for publication and agreed to be held accountable for the work performed therein.

Competing interests. We declare we have no competing interests.

Funding. The work was funded by the National Science Foundation (EAR 1725154) and Vanderbilt University.

Acknowledgments. We thank the Florida Museum of Natural History for allowing the collection of new data (B. MacFadden and J. Bloch) and J. Curtis (University of Florida) for analytical assistance. We are grateful to the faculty, staff, and students at Vanderbilt University and numerous researchers for conversations regarding these data, including the reviews of anonymous reviewers who provided constructive feedback.

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
