## [Peer Review File · Proceedings of the Royal Society B: Biological Sciences]

Review History

RSPB-2021-1839.R0 (Original submission)

Review form: Reviewer 1

Recommendation

Major revision is needed (please make suggestions in comments)

Scientific importance: Is the manuscript an original and important contribution to its field?

Good

General interest: Is the paper of sufficient general interest?

Good

Quality of the paper: Is the overall quality of the paper suitable?

Acceptable

Is the length of the paper justified?

Yes

Should the paper be seen by a specialist statistical reviewer?

No

Do you have any concerns about statistical analyses in this paper? If so, please specify them explicitly in your report.

No

It is a condition of publication that authors make their supporting data, code and materials available - either as supplementary material or hosted in an external repository. Please rate, if applicable, the supporting data on the following criteria.

Is it accessible?

Yes

Is it clear?

Yes

Is it adequate?

Yes

Do you have any ethical concerns with this paper?

No

Comments to the Author

Please see my comments in the attached PDF (Appendix A).

Review form: Reviewer 2

Recommendation

Accept with minor revision (please list in comments)

Scientific importance: Is the manuscript an original and important contribution to its field?

Acceptable

General interest: Is the paper of sufficient general interest?

Good

Quality of the paper: Is the overall quality of the paper suitable?

Good

Is the length of the paper justified?

Yes

Should the paper be seen by a specialist statistical reviewer?

No

Do you have any concerns about statistical analyses in this paper? If so, please specify them explicitly in your report.

No

It is a condition of publication that authors make their supporting data, code and materials available - either as supplementary material or hosted in an external repository. Please rate, if applicable, the supporting data on the following criteria.

Is it accessible?

Yes

Is it clear?

Yes

Is it adequate?

Yes

Do you have any ethical concerns with this paper?

No

Comments to the Author

Review of RSPB-2021-1839 "Global long-term stability of individual dietary specialization in herbivorous mammals" by DeSantis et al.

This manuscript analyzes a dataset of stable carbon isotope serial samples from mammalian herbivore dentitions to ask whether 'dietary generalism' in a species emerges from intra- or inter-individual dietary variation. They find that most generalist species are comprised of individual specialists, with grazers showing the highest levels of individual variation (based on the authors' isotopic individuality index).

This is an interesting ecological question, but it has been addressed in previous work as noted in the introduction. Nonetheless, this study's synthesis of existing stable isotope data in the literature and inclusion of new samples from C3-C4 mixed-feeding taxa is a nice addition to the existing literature. I have a few comments/questions for the authors to consider.

Major Comments

Isotopes are a proxy for diet and record dietary information over specific time scales. There is surprisingly no discussion of how dental maturation rates across taxa relate to serial samples, given the fact that sampling rates and distances vary, in addition to fundamental differences in the dental biology of taxa analyzed. The authors consider variation in crown height (and use a five-sample sliding window in an attempt to control for this) but should also discuss how variation in the timing of the individual's life is reflected in serial samples – Are all samples considered to be annual, semi-annual? Does this vary across taxa? Does it matter? Without these considerations, the manuscript feels incomplete.

I ask these questions because depending on the scale of diet information recorded, you could have a scenario where a dietary generalist looks like a specialist (from the view of isotopes) because of temporal averaging of isotope signals. We all acknowledge that bulk samples record average diet, but even serial samples are averaging diet over a certain span of time. If that span of time systematically varies across taxa, that adds a major complication. For example, could greater intratooth $\delta^{13}C$ range in grazers reflect faster dental maturation times due to taller crowns and thus less temporal averaging of isotopic variability?

How does consideration of variation in plant isotopic composition impact these conclusions? For example, Cerling and others have discussed differences between NADP and NAD+PEP-CK grasses in the African context. Could systematic differences in isotopic variability between browse and graze impact your findings?

Minor Comments

pg 7, line 217-220: But what about time in terms of the osteological collections? A lot of museums house collections that were collected across several decades, if not centuries, and probably sample sporadically across a species' geographic distribution. Could species dietary niches shift on those scales?

pg 9, line 382: "Individuals identified to the genus level were lumped with congener individuals identified to the species level when it could be reasonably inferred that they belonged to the same

species (e.g., *Mammuthus* sp. lumped with *Mammuthus columbi*)" –does this vary by feeding group? Could unidentified species be inflating genus level variation?

Decision letter (RSPB-2021-1839.R0)

18-Oct-2021

Dear Dr DeSantis:

Your manuscript has now been peer reviewed and the reviews have been assessed by an Associate Editor. The reviewers' comments (not including confidential comments to the Editor) and the comments from the Associate Editor are included at the end of this email for your reference. As you will see, the reviewers and the Editors have raised some concerns with your manuscript and we would like to invite you to revise your manuscript to address them.

Research ethics:

Use of animals and field studies:

It is a condition of publication that you make available the data and research materials supporting the results in the article. Please see our Data Sharing Policies (<https://royalsociety.org/journals/authors/author-guidelines/#data>). Datasets should be deposited in an appropriate publicly available repository and details of the associated accession number, link or DOI to the datasets must be included in the Data Accessibility section of the article (<https://royalsociety.org/journals/ethics-policies/data-sharing-mining/>). Reference(s) to datasets should also be included in the reference list of the article with DOIs (where available).

Please submit a copy of your revised paper within three weeks. If we do not hear from you within this time your manuscript will be rejected. If you are unable to meet this deadline please let us know as soon as possible, as we may be able to grant a short extension.

Best wishes,

Dr John Hutchinson, Editor

Associate Editor

Comments to Author:

Thank you for your submission. Both reviewers agree that the dataset is strong and that this will make a positive contribution to the literature, although they both have concerns that will need to be addressed before the paper can be accepted. These include Reviewer 1's concerns regarding the pre-assignment of species into body size groups prior to statistical comparisons – additional justification of this approach will be necessary. Also, Reviewer 2 suggests that greater care be taken to place these conclusions within the context of prior work, as the main conclusions are not entirely novel.

The statistical methods are relatively straightforward and generally adequate. My only concern with their analyses are not with the stats itself, but with the pre-assignment of species into body size groups before making statistical comparisons among them. The assignments appear to be somewhat arbitrary, and I think some justifications (perhaps in the supplementary material) would be needed.

Reviewer(s)' Comments to Author:

Referee: 1

Comments to the Author(s)

Please see my comments in the attached PDF.

Referee: 2

Comments to the Author(s)

Review of RSPB-2021-1839 "Global long-term stability of individual dietary specialization in herbivorous mammals" by DeSantis et al.

This manuscript analyzes a dataset of stable carbon isotope serial samples from mammalian herbivore dentitions to ask whether 'dietary generalism' in a species emerges from intra- or inter-individual dietary variation. They find that most generalist species are comprised of individual specialists, with grazers showing the highest levels of individual variation (based on the authors' isotopic individuality index).

This is an interesting ecological question, but it has been addressed in previous work as noted in the introduction. Nonetheless, this study's synthesis of existing stable isotope data in the literature and inclusion of new samples from C3-C4 mixed-feeding taxa is a nice addition to the existing literature. I have a few comments/questions for the authors to consider.

Major Comments

Isotopes are a proxy for diet and record dietary information over specific time scales. There is surprisingly no discussion of how dental maturation rates across taxa relate to serial samples, given the fact that sampling rates and distances vary, in addition to fundamental differences in the dental biology of taxa analyzed. The authors consider variation in crown height (and use a five-sample sliding window in an attempt to control for this) but should also discuss how variation in the timing of the individual's life is reflected in serial samples – Are all samples considered to be annual, semi-annual? Does this vary across taxa? Does it matter? Without these considerations, the manuscript feels incomplete.

I ask these questions because depending on the scale of diet information recorded, you could have a scenario where a dietary generalist looks like a specialist (from the view of isotopes) because of temporal averaging of isotope signals. We all acknowledge that bulk samples record average diet, but even serial samples are averaging diet over a certain span of time. If that span of time systematically varies across taxa, that adds a major complication. For example, could greater intratooth $\delta^{13}\text{C}$ range in grazers reflect faster dental maturation times due to taller crowns and thus less temporal averaging of isotopic variability?

How does consideration of variation in plant isotopic composition impact these conclusions? For example, Cerling and others have discussed differences between NADP and NAD+PEP-CK grasses in the African context. Could systematic differences in isotopic variability between browse and graze impact your findings?

Minor Comments

pg 7, line 217-220: But what about time in terms of the osteological collections? A lot of museums house collections that were collected across several decades, if not centuries, and probably sample sporadically across a species' geographic distribution. Could species dietary niches shift on those scales?

pg 9, line 382: "Individuals identified to the genus level were lumped with congener individuals identified to the species level when it could be reasonably inferred that they belonged to the same species (e.g., *Mammuthus* sp. lumped with *Mammuthus columbi*)" -does this vary by feeding group? Could unidentified species be inflating genus level variation?

Author's Response to Decision Letter for (RSPB-2021-1839.R0)

See Appendix B.

Decision letter (RSPB-2021-1839.R1)

10-Jan-2022

Dear Dr DeSantis

I am pleased to inform you that your manuscript entitled "Global long-term stability of individual dietary specialization in herbivorous mammals" has been accepted for publication in Proceedings B. Congratulations!!

Data Accessibility section

Open Access

You are invited to opt for Open Access, making your freely available to all as soon as it is ready for publication under a CCBY licence. Our article processing charge for Open Access is £1700. Corresponding authors from member institutions (<http://royalsocietypublishing.org/site/librarians/allmembers.xhtml>) receive a 25% discount to these charges. For more information please visit <http://royalsocietypublishing.org/open-access>.

Paper charges

Sincerely,

Dr John Hutchinson

Associate Editor:

Board Member

Comments to Author:

The current version of the manuscript is improved, and does a better job placing the current study in the context of prior work. The dataset assembled here is impressive and will influence the future direction of palaeoecological, and present-day ecological studies on intraspecific variation in diet.

Appendix A

General comments:

This study provides insights into the topics of niche theories, isotope ecology, and dietary evolution. The compilation of published serial sample data, along with newly collected data, are a valuable contribution to the field. This well-curated database is potentially useful for many future studies.

The paper investigates interesting questions about individual specialization/generalization within generalist species. The findings reveal an intriguing new phenomenon of the prevalence of individual specialists across dietary groups, shedding light into the previously lesser-known niche of “generalist specialist.”

While the study poses important research questions and has sufficient raw data for addressing the questions, I think there is still room for improvements, particularly on the following aspects:

1. I strongly recommend the authors to give more considerations to the dampening effect of the isotopic signature recorded in tooth enamel. Although this has been mentioned in the paper, the authors have not provided convincing enough evidence for why one should not be concerned about it. While I do not think this effect necessarily changes the overall conclusion of this paper (that individual specialization is a widespread phenomenon, or is at least more common than previously recognized), I do think more passages are needed to acknowledge the potential influence of the dampening of isotopic signature. Only when an individual consumes one type of plant for weeks/months in a row and then shifts to a different diet for subsequent weeks/months can the isotopic variability of its diet be faithfully recorded along the growth axis of its teeth. If an individual consumes a wide range of vegetation but does not shift its diet much seasonally, what is recorded would be the overall isotopic signature of the ingested plants, which does not fully capture the carbon isotope values and the relative proportions of the various types of plants consumed.
2. Low $\delta^{13}C$ in mixed feeders is primarily due to high species-level variability. Because generalist species are typically more wide-ranging than specialist species, the intraspecific variation can be a direct reflection of geographic range and vegetation availability. In other words, individual specialization within a given generalist species could reflect reduced local competition as this paper has discussed, or it could simply reflect geographic variation in plant availability and has little to do with intra- or inter-specific competition. I recommend that the authors give more consideration to the “space” element (i.e. how geographic variation in plant availability affect species-level variability and $\delta^{13}C$ index), especially when there are statements in the paper about how the pattern is common across space and time.
3. In addition to differentiating samples from below 37 deg latitude, it may be worth re-running some of the analyses for specimens that post-date the expansion of C_4 vegetation in their respective continents.
4. I think it would be worth mentioning the range of body masses of the sampled taxa before giving the body mass bins. Because the relationship between herbivore body size and dietary ecology is rather complicated, I think if an evaluation of the effect of body size is to be done, it needs to be done carefully. Brief justifications for how boundaries between the body size bins are drawn are necessary—as of now, 100 kg and 350 kg seem to be arbitrary. Ideally, the boundary values should be of some biological or physiological relevance. If that

is the case, it needs to be explained. If the divisions are made primarily for statistical considerations (i.e. to have comparable numbers of individuals in small, medium, and large groups), this needs to be stated too.

5. Even though the mean individual ranges are generally narrow, grazers' range is ~1.6 times as wide as browsers' (and 2.4 times for <37 deg latitude), and the ranges are also significantly different between browsers and grazers (Table 1). You should talk about what this range can mean, in terms of the vegetation consumed. This would be helpful to non-specialist readers.

Additionally, the paper would also use more consistent terminology, particularly when it comes to describing dietary groups. Some in-text references to figures are incorrect—not all figures cited show what the text is talking about. Figure captions and axis labels could also be more accurate and informative. The Introduction and Discussion sections will also need some re-writing.

Detailed comments with references to line numbers:

“enamel” should be subscripted

maybe specify what you mean by dietary strategy by listing the three dietary groups in parentheses?

40–43 the example of *C. emsliei* would probably go better with the previous sentence—that “almost all herbivores...are composed of individual specialists,” as it explains to readers clearly what you mean by individual specialists—than with the statement that this pattern holds true through time, as there is only one age associated with this example

“define”

add “choice” after “habitat”, as diet does not directly influence the physical environment per se. The rest of the sentence also needs some re-writing for better clarity. For example, I would how animals move in their environment is equivalent to migration, while migration is one aspect of landscape use and should not be mixed up with the latter.

Before talking about humans, say a little more about how animals' dietary niche, especially how niche breadths have typically been defined as either broad or narrow for a certain species, then transition into how this new niche is unique

omit “for example”

69–71 this sentence (“In fact...”) seems to imply that such investigations have been previously done many times and revealed robust patterns

thought to be or found to be?

move “dietary” to before “specialist” to make the sentence flow better

76–81 This paragraph seems redundant to me. Basically, it is saying with examples that some generalist species are composed of generalist individuals (thereby supporting the NVH) while

many other species specialist individuals (thereby not supporting the NVH). Since this has already been stated in the previous paragraph, and carnivores are not the focus of this study, I recommend keeping only the last sentence of this paragraph.

before the term mixed-feeders appears here (for the first time), it is worth introducing your three dietary groups first and briefly explaining the conventional view of their dietary niche breadths

the duration of tooth formation is relevant and important enough here that I think there is no need for parenthesis

109–113 I recommend removing the topics of “shifts in plant species composition” and “species distributions through time” from Introduction since you did not revisit these topics in Discussion

I cannot tell from Figs. 1–2 that there is as much as an 8‰ range in grazers. Only Fig. 3 shows that.

125–126 What’s in parenthesis here goes better after the first sentence of the paragraph where you report the total range of carbon isotope values. Also, is it necessary to list all the countries here? Perhaps it would be better to include a map of sample locations in the supplementary file.

“herbivores”

what sampling standardization?

166–168 by “between” I think you mean “among”, since the comparisons involve more than three values

by “among” I think you mean “in”

Again, Fig. 4 does not show individual variabilities, but averaged individual variabilities. The 8‰ range mentioned in this sentence is not shown in Fig. 4.

188–189 morphologically “and nutritionally” constrained; would also be good to include citations here about browsing species’ constraints

190–191 I’d be curious to see what their serial sample profiles look like, perhaps it would be worth graphing them (like what is done for *C. embliei* in Fig. 2) in the supplementary file

200–216 The argument in this paragraph is relatively weak. The fact that different parts of grass forage do not get selected upon as finely as the architectural heterogeneous browse materials would contribute to higher isotopic variability in grazers only if the different parts of grasses have highly variable isotopic signatures, and I’m not sure if that’s the case. I think how feeding selectivity translates to isotopic variability is a little more complex than what is described here. Note that there is greater variability in the C₃ plants than in C₄ plants, and when grasses are consumed as a whole, only one overall isotopic signal is recorded in the teeth regardless of how much variability is among the plant parts. Additionally, the second sentence (tradeoffs for consuming grasses) and the last sentence (mixed feeders having the lowest III indices) of this paragraph seem out of place. Their logical linkage to the rest of the paragraph is not strong. You need to write elaborate on both statements if you keep them where they are.

217–218 While time averaging does not affect what you found for individual teeth (Figs.1-3), you may want to consider how it may affect your calculated III indices (Fig. 4). In other words, the III values of extinct taxa are potentially lowered by the effect of time averaging.

223 this “example” does not match what you just talked about...it is critical in this paper to address the dampening effect of recorded isotopic signatures, and to make a sound argument here you need examples of a single tooth recording a wide range of carbon isotope values

I am a little bothered by “there is no reason why...”, as you are trying to give an example to support an argument. The absence of evidence is not the evidence of absence.

227–228 Sentence in parenthesis is a little redundant

what roles?

243–247 These two sentences are a bit confusing. First, you talk about individuals consuming disparate foods, which to me implies more generalist individuals, then you talk about individual specialization in the second sentence. Perhaps you can modify the phrasing in the first sentence.

what do you mean by “extreme”? is there a reason to emphasize on extreme generalists (as opposed to other, less extreme generalists) here?

277 Fig. 4 does not indicate age, nor are taxon ages noted in the supplement

Comments on figures and tables:

In all figures the Y-axis label should include “enamel” to match the text, and the unit should be in parenthesis, making it: $\delta^{13}\text{C}_{\text{enamel}} (\text{‰})$

Figure 1

554 “breadths”

555 “if individuals have more specialized (left) or more generalized (right) diets”

This figure is very intriguing and informative, with theoretical predictions (left and middle columns) and empirical results (right column) shown simultaneously. It is probably the one figure that best summarizes the key findings of this study, and so it would be good to improve it further. I have three suggestions: (1) Symbols in figure legend should match the colors in the graph, since each symbol (species) only has one corresponding dietary assignment (color). (2) In the figure caption, it would be worth noting that tooth crown height generally increases with higher levels of grazing. (3) It is not clear to me why these six taxa are chosen as examples to be presented here. Why not illustrate the entirety of your dataset?

Figure 2

563 “(range = 13.4‰)”

564–565 just say “browsers” and “grazers” since (1) “obligate” is not defined or used elsewhere in the paper, and (2) in the literature, obligate grazers are distinguished from variable grazers and have quantifiable defining criteria, so do not mix up grazers with obligate grazers if what you really mean here is grazers

Figure 3

569–570 end members of a range should have the same number of decimal places

570 not clear to me what you mean by “methods”

For each panel, give the sample size (numbers of individuals) of the dataset ($n = \#\#$). Avoid using “trophic group” in Y-axis labels, since you don’t use these words anywhere else in the paper. Just say “dietary category.” I would also remove “Individual Isotopic Variability” from the panel headings, since they are all the same, and incorporate the phrase into figure caption if necessary.

Figure 4

Panels A) and C) need more informative Y-axis labels (add “species” to Y axis to contrast the “average individual” on X axis)

I don’t really see what you describe about panel C)—that species in low latitudes plot close to the 1:1 line than other species.

Figure caption should include explanation for the box and whisker plots; alternatively, add an inset diagram to panel B) or D).

Table 1

To my understanding, “n” denotes number of individuals for range (top half of the table) and number of species for III (bottom half of the table). If so, the abbreviation explanation should be corrected at the bottom of the table.

6 December 2021

RE: Global long-term stability of individual dietary specialization in herbivorous mammals

By L.R.G. DeSantis, M.I. Pardi, A. Du, M.A. Greshko, L.T. Yann, R.C. Hulbert, Jr., J. Louys

Dear Associate Editor John Hutchinson:

Thank you for your consideration of this manuscript. Based on the reviews, all reviewers have commented on the importance of this data set and the conclusions drawn from these data. Two primary comments were made: 1) the need to address the body size categories used (i.e., why these categories were used prior to a statistical analysis), and 2) the need to better link the conclusions of this paper with prior work. We appreciate these comments and agree that an improved discussion of both topics is important. We have made both broad and specific changes to address these and other line-item comments. We are appreciative of each of the reviewers who took the time during the COVID-19 pandemic to review this manuscript. We appreciate all their comments and feedback, and believe the resulting manuscript is stronger because of their reviews.

Reviewer #1: This study provides insights into the topics of niche theories, isotope ecology, and dietary evolution. The compilation of published serial sample data, along with newly collected data, are **a valuable contribution to the field. This well-curated database is potentially useful for many future studies.** The paper investigates interesting questions about individual specialization/generalization within generalist species. **The findings reveal an intriguing new phenomenon of the prevalence of individual specialists across dietary groups, shedding light into the previously lesser-known niche of “generalist specialist.”**

We thank the reviewer for their comments regarding the importance and value of this contribution.

While the study poses important research questions and has sufficient raw data for addressing the questions, I think there is still room for improvements, particularly on the following aspects:

1. I strongly recommend the authors to give more considerations to the dampening effect of the isotopic signature recorded in tooth enamel. Although this has been mentioned in the paper, the authors have not provided convincing enough evidence for why one should not be concerned about it. While I do not think this effect necessarily changes the overall conclusion of this paper (that individual specialization is a widespread phenomenon, or is at least more common than previously recognized), I do think more passages are needed to acknowledge the potential influence of the dampening of isotopic signature.

Only when an individual consumes one type of plant for weeks/months in a row and then shifts to a different diet for subsequent weeks/months can the isotopic variability of its diet be faithfully recorded along the growth axis of its teeth. If an individual consumes a wide range of vegetation but does not shift its diet much seasonally, what is recorded would be the overall isotopic signature of the ingested plants, which does not fully capture the carbon isotope values and the relative proportions of the various types of plants consumed.

We have now expanded the discussion of the dampening effect of the isotopic signature to specifically mention instances when an animal's averaged diet is not fully captured if the averaged values of different sources are in similar proportion throughout time. See discussion.

2. Low III in mixed feeders is primarily due to high species-level variability. Because generalist species are typically more wide-ranging than specialist species, the intraspecific variation can be a direct reflection of geographic range and vegetation availability. In other words, individual specialization within a given generalist species could reflect reduced local competition as this paper has discussed, or it could simply reflect geographic variation in plant availability and has little to do with intra- or inter-specific competition. I recommend that the authors give more consideration to the “space” element (i.e. how geographic variation in plant availability affect species-level variability and III index), especially when there are statements in the paper about how the pattern is common across space and time.

This is a good point that we now have added to the discussion, explicitly the discussion of geographic variation in plant availability. We also do want to reiterate that this is one of the reasons we added analysis of only low-latitude specimens, so that individuals sampled occur in places where isotopic variability is possible (though of course some regional variation and potential for movement will exist, there is the potential for C3 and C4 vegetation to be consumed). See discussion.

3. In addition to differentiating samples from below 37 deg latitude, it may be worth re-running some of the analyses for specimens that post-date the expansion of C4 vegetation in their respective continents.

Despite the earlier innovation of C4 vegetation, the expansion occurred largely between 5-7 million years. Regions where it was delayed include Australia, parts of Asia (e.g. the Tibetan plateau), and large portions of Europe (where C4 grasses never became dominant, in large part due to occurring at higher latitudes that experience cooler climates). While we placed our data cutoff for inclusion during the Miocene to maximize the number of samples possible, we now do note the percentage of specimens that occur at during the early Miocene prior to the expansion of C4 grasses in their region. In regions where C4 expansion was delayed (e.g., Australasia, the Tibetan plateau), all serial samples occur during the Pliocene to late Pleistocene (well within the time frame where C4 plants had expanded). Further, in most of the studies from which data was gathered, the aim of serial sampling of tooth enamel was to infer the degree to which herbivores consumed C3 and/or C4 vegetation (with only a minority of studies conducted in primarily C3 ecosystems). We have clarified this in the methods section.

4. I think it would be worth mentioning the range of body masses of the sampled taxa before giving the body mass bins. Because the relationship between herbivore body size and dietary ecology is rather complicated, I think if an evaluation of the effect of body size is to be done, it needs to be done carefully. Brief justifications for how boundaries between the body size bins are drawn are necessary—as of now, 100 kg and 350 kg seem to be arbitrary. Ideally, the boundary values should be of some biological or physiological relevance. If that is the case, it needs to be explained. If the divisions are made primarily for statistical considerations (i.e. to have comparable numbers of individuals in small, medium, and large groups), this needs to be stated too.

We used the body masses categories of Bunn 1982, Brain 1983, and commonly used by hundreds of others included Faith et al. 2019. We have clarified this in the methods section.

5. Even though the mean individual ranges are generally narrow, grazers' range is ~1.6 times as wide as browsers' (and 2.4 times for <37 deg latitude), and the ranges are also significantly different between browsers and grazers (Table 1). You should talk about what this range can mean, in terms of the vegetation consumed. This would be helpful to nonspecialist readers.

We agree and have expanded the discussion of the $\delta^{13}\text{C}$ ranges of browsers and grazers, in terms of vegetation consumed.

Minor Comments:

Additionally, the paper would also use more consistent terminology, particularly when it comes to describing dietary groups. Some in-text references to figures are incorrect—not all figures cited show what the text is talking about. Figure captions and axis labels could also be more accurate and informative. The Introduction and Discussion sections will also need some rewriting. Detailed comments with references to line numbers:

“enamel” should be subscripted

Done

maybe specify what you mean by dietary strategy by listing the three dietary groups in parentheses?

Done. We have added what we mean by dietary strategy (i.e., browser, mixed-feeder, grazer).

40–43 the example of *C. emsliei* would probably go better with the previous sentence—that“almost all herbivores...are composed of individual specialists,” as it explains to readers clearly what you mean by individual specialists—than with the statement that this pattern holds true through time, as there is only one age associated with this example

*Great point, we have moved this up and now moved discussion of time to later in the abstract. We now state: “For example, *Cormohipparion emsliei* (Equidae) from the Pliocene of Florida (~5 Ma) exhibit a $\delta^{13}\text{C}_{\text{enamel}}$ range of 13.4‰, but all individuals sampled have $\delta^{13}\text{C}_{\text{enamel}}$ ranges of $\leq 2\%$ (mean = 1.1‰). Most notably, this pattern holds globally and through time, with almost all herbivorous mammal individuals exhibiting narrow $\delta^{13}\text{C}_{\text{enamel}}$ ranges ($\leq 3\%$), demonstrating that individuals are specialized and less representative of their overall species dietary breadth.*

“define”

Corrected.

add “choice” after “habitat”, as diet does not directly influence the physical environment per se. The rest of the sentence also needs some re-writing for better clarity. For example, I would how animals move in their environment is equivalent to migration, while migration is one aspect of landscape use and should not be mixed up with the latter.

Done. We have added choice after habitat, but have also clarified what we mean by “how animals move in their environment” as we literally mean how animals move (e.g., run, climb). We now state, “Diet influences an animal's habitat choice, landscape use/migration, how animals move in their environment (i.e., their biomechanics as pertains to food acquisition), and even reproduction (1–4)”.

Before talking about humans, say a little more about how animals' dietary niche, especially how niche breadths have typically been defined as either broad or narrow for a certain species, then transition into how this new niche is unique

Done.

omit “for example”

Done.

69–71 this sentence (“In fact...”) seems to imply that such investigations have been previously done many times and revealed robust patterns

We removed “In fact” and added “—though only a handful of species have been examined.”

thought to be or found to be?

We removed this paragraph as suggested by this reviewer; no change is needed.

move “dietary” to before “specialist” to make the sentence flow better

Done.

76–81 This paragraph seems redundant to me. Basically, it is saying with examples that some generalist species are composed of generalist individuals (thereby supporting the NVH) while many other species specialist individuals (thereby not supporting the NVH). Since this has already been stated in the previous paragraph, and carnivores are not the focus of this study, I recommend keeping only the last sentence of this paragraph.

Done.

before the term mixed-feeders appears here (for the first time), it is worth introducing your three dietary groups first and briefly explaining the conventional view of their dietary niche breadths

Great point, we have now done this and the sentences read as follows: “Here, we use stable carbon isotope data from serial samples of mammal tooth enamel ($\delta^{13}\text{C}_{\text{enamel}}$) collected from herbivores that can be categorized into the dietary groups of grazer, browser, and mixed feeder (i.e., eating both grass and browse). We gathered data from the literature dating from the Miocene to the present (298 individuals, 4013 samples) and acquired new samples of mixed-feeders (20 individuals, 121 samples).

the duration of tooth formation is relevant and important enough here that I think there is no need for parenthesis

Parentheses were removed.

109–113 I recommend removing the topics of “shifts in plant species composition” and “species distributions through time” from Introduction since you did not revisit these topics in Discussion

Done.

I cannot tell from Figs. 1–2 that there is as much as an 8‰ range in grazers. Only Fig. 3 shows that.

Corrected.

125–126 What’s in parenthesis here goes better after the first sentence of the paragraph where you report the total range of carbon isotope values. Also, is it necessary to list all the countries here? Perhaps it would be better to include a map of sample locations in the supplementary file.

The full list is more extensive, these were just some examples; however, all countries are listed in the supplemental data table. We now clarify this and moved this statement to be a part of the first sentence, as suggested.

“herbivores”

Corrected.

what sampling standardization?

We have clarified this, by stating (i.e., the sliding window analysis).

166–168 by “between” I think you mean “among”, since the comparisons involve more than three values

Corrected.

by “among” I think you mean “in”

Corrected.

Again, Fig. 4 does not show individual variabilities, but averaged individual variabilities. The 8‰ range mentioned in this sentence is not shown in Fig. 4.

Corrected.

188–189 morphologically “and nutritionally” constrained; would also be good to include citations here about browsing species’ constraints

Corrected.

190–191 I’d be curious to see what their serial sample profiles look like, perhaps it would be worth graphing them (like what is done for *C. embliei* in Fig. 2) in the supplementary file.

This is a great suggestion, and we are happy to add this to the supplementary files (now Figure S3).

200–216 The argument in this paragraph is relatively weak. The fact that different parts of grass forage do not get selected upon as finely as the architectural heterogeneous browse materials would contribute to higher isotopic variability in grazers only if the different parts of grasses have highly variable isotopic signatures, and I'm not sure if that's the case. I think how feeding selectivity translates to isotopic variability is a little more complex than what is described here.

Note that there is greater variability in the C3 plants than in C4 plants, and when grasses are consumed as a whole, only one overall isotopic signal is recorded in the teeth regardless of how much variability is among the plant parts. Additionally, the second sentence (tradeoffs for consuming grasses) and the last sentence (mixed feeders having the lowest III indices) of this paragraph seem out of place. Their logical linkage to the rest of the paragraph is not strong. You need to write elaborate on both statements if you keep them where they are.

We have reorganized and largely rewritten this paragraph to make the following points: 1) $\delta^{13}\text{C}_{\text{enamel}}$ values can vary spatially based on habitat heterogeneity and selection of plant parts by herbivores, 2) the high variability in grazers, while potentially stemming from spatial variability is unlikely stemming from isotopic variability within grasses, due to challenges of foraging on discreet plant parts with different isotopic values.

217–218 While time averaging does not affect what you found for individual teeth (Figs.1-3), you may want to consider how it may affect your calculated III indices (Fig. 4). In other words, the III values of extinct taxa are potentially lowered by the effect of time averaging.

We have now included discussion of how III indices pertain to extant vs. extinct taxa.

this “example” does not match what you just talked about...it is critical in this paper to address the dampening effect of recorded isotopic signatures, and to make a sound argument here you need examples of a single tooth recording a wide range of carbon isotope values

We have changed the example here to instead focus on the fact that a single tooth can have high variability; thus, while it is possible, it is just rare. See discussion.

I am a little bothered by “there is no reason why...”, as you are trying to give an example to support an argument. The absence of evidence is not the evidence of absence.

This has been removed, as a different example is used to support the statements in the paragraph.

227–228 Sentence in parenthesis is a little redundant

This has been removed, as a different example is used to support the statements in the paragraph.

what roles?

We have now stated, “Individual foraging behavior can determine how other members of a foraging group behave, a group’s choice of where to forage, and foraging methods and their successes (48–51), while individual temperament can affect predator-prey interactions (52).”

243–247 These two sentences are a bit confusing. First, you talk about individuals consuming disparate foods, which to me implies more generalist individuals, then you talk about individual specialization in the second sentence. Perhaps you can modify the phrasing in the first sentence.

We have modified the phrasing in the first sentence.

what do you mean by “extreme”? is there a reason to emphasize on extreme generalists (as opposed to other, less extreme generalists) here?

We have removed the word extreme as it doesn’t change the meaning of our sentence. Good point!

Fig. 4 does not indicate age, nor are taxon ages noted in the supplement

We have reworded this statement.

Comments on figures and tables:

In all figures the Y-axis label should include “enamel” to match the text, and the unit should be in parenthesis, making it: $\delta^{13}\text{C}_{\text{enamel}}$ (‰)

Corrected.

Figure 1

554 “breadths”

Corrected

555 “if individuals have more specialized (left) or more generalized (right) diets”

This figure is very intriguing and informative, with theoretical predictions (left and middle columns) and empirical results (right column) shown simultaneously. It is probably the one figure that best summarizes the key findings of this study, and so it would be good to improve it further. I have three suggestions: (1) Symbols in figure legend should match the colors in the graph, since each symbol (species) only has one corresponding dietary assignment (color). (2) In the figure caption, it would be worth noting that tooth crown height generally increases with higher levels of grazing. (3) It is not clear to me why these six taxa are chosen as examples to be presented here. Why not illustrate the entirety of your dataset?

We have only provided a subset of these data to illustrate these points (as examples), due to the need for visual clarity. If we had plotted all data, it would not be possible to make sense of any of these data. Please note that we summarize all data in Fig 3 and in all subsequent analyses. Further, we have also added examples with the highest variability to Figure S3.

Figure 2

563 “(range = 13.4%)”

Corrected

564–565 just say “browsers” and “grazers” since (1) “obligate” is not defined or used elsewhere in the paper, and (2) in the literature, obligate grazers are distinguished from variable grazers and have quantifiable defining criteria, so do not mix up grazers with obligate grazers if what you really mean here is grazers

We have removed the word obligate here so as not to not confuse the reader. Thanks!

Figure 3

569–570 end members of a range should have the same number of decimal places

Corrected

570 not clear to me what you mean by “methods”

For each panel, give the sample size (numbers of individuals) of the dataset ($n = ##$). Avoid using “trophic group” in Y-axis labels, since you don’t use these words anywhere else in the paper. Just say “dietary category.” I would also remove “Individual Isotopic Variability” from the panel headings, since they are all the same, and incorporate the phrase into figure caption if necessary.

We have changed this to, “when calculated using the full range of statistical methods.”

Figure 4

Panels A) and C) need more informative Y-axis labels (add “species” to Y axis to contrast the “average individual” on X axis)

I don’t really see what you describe about panel C)—that species in low latitudes plot close to the 1:1 line than other species.

Figure caption should include explanation for the box and whisker plots; alternatively, add an inset diagram to panel B) or D).

We have made this change to panels A and C.

Table 1

To my understanding, “n” denotes number of individuals for range (top half of the table) and number of species for III (bottom half of the table). If so, the abbreviation explanation should be corrected at the bottom of the table.

Corrected

Reviewer #2: This manuscript analyzes a dataset of stable carbon isotope serial samples from mammalian herbivore dentitions to ask whether ‘dietary generalism’ in a species emerges from intra- or inter- individual dietary variation. They find that most generalist species are comprised of individual specialists, with grazers showing the highest levels of individual variation (based on the authors’ isotopic individuality index).

This is an interesting ecological question, but it has been addressed in previous work as noted in the introduction. Nonetheless, this study’s synthesis of existing stable isotope data in the literature and inclusion of new samples from C3-C4 mixed-feeding taxa is a nice addition to the existing literature. I have a few comments/questions for the authors to consider.

We thank the reviewer and do want to make clear in this paper that we are building off the work and ideas of others and have tried to make this clearer in the discussion and conclusion, adding in reference to those earlier cited papers in the discussion.

Major Comments

Isotopes are a proxy for diet and record dietary information over specific time scales. There is surprisingly no discussion of how dental maturation rates across taxa relate to serial samples, given the fact that sampling rates and distances vary, in addition to fundamental differences in the dental biology of taxa analyzed. The authors consider variation in crown height (and use a five-sample sliding window in an attempt to control for this) but should also discuss how variation in the timing of the individual’s life is reflected in serial samples—Are all samples considered to be annual, semi-annual? Does this vary across taxa? Does it matter? Without these considerations, the manuscript feels incomplete.

I ask these questions because depending on the scale of diet information recorded, you could have a scenario where a dietary generalist looks like a specialist (from the view of isotopes) because of temporal averaging of isotope signals. We all acknowledge that bulk samples record average diet, but even serial samples are averaging diet over a certain span of time. If that span of time

systematically varies across taxa, that adds a major complication. For example, could greater intratooth $\delta^{13}\text{C}$ range in grazers reflect faster dental maturation times due to taller crowns and thus less temporal averaging of isotopic variability?

We thank the reviewer for pointing out these omissions. As most of these studies were done with the aim of being largely comparable to one another, the effect of temporal averaging is fairly similar across taxa. That being said, discussion of this has been added to the discussion section.

How does consideration of variation in plant isotopic composition impact these conclusions? For example, Cerling and others have discussed differences between NADP and NAD+PEP-CK grasses in the African context. Could systematic differences in isotopic variability between browse and graze impact your findings?

We don't believe differences in NADP and NAD+PEP-CK grasses would contribute significant variability in grasses, as the difference between these sub-pathways is only ~1%. We now cite their work and mention this in the discussion.

Minor Comments

pg 7, line 217-220: But what about time in terms of the osteological collections? A lot of museums house collections that were collected across several decades, if not centuries, and probably sample sporadically across a species' geographic distribution. Could species dietary niches shift on those scales?

While it is possible for animals to switch diets over a period of years to decades, each analysis is focused on the duration of time during which a tooth is mineralized, so ~1 year. We don't discount this possibility for longer term change and have added discussion of this to the discussion.

pg 9, line 382: "Individuals identified to the genus level were lumped with congener individuals identified to the species level when it could be reasonably inferred that they belonged to the same species (e.g., *Mammuthus* sp. lumped with *Mammuthus columbi*)" –does this vary by feeding group? Could unidentified species be inflating genus level variation?

*We changed this to: "We only analyzed those teeth with at least five serial samples. Individuals identified to the genus level were lumped with congener individuals identified to the species level only when it could be reasonably inferred based on location and geological time period that they belonged to the same species (i.e., *Mammuthus* sp. lumped with *Mammuthus columbi*)." We only did this for *Mammuthus* based on expert consensus that this is the case.*

Summary

Collectively, this manuscript has been well reviewed by all reviewers and we were able to address all of the reviewer comments and suggestions in the resulting manuscript. Included is a track-changed version of the manuscript. The manuscript contains four color figures and supplementary information (we have added one supplemental figure, based on feedback from Reviewer 1). The corresponding author's contact information for communication regarding this manuscript is stated above (e-mail: larisa.desantis@vanderbilt.edu).

Thank you for your continued consideration of the manuscript.

Sincerely,

Larisa R. G. DeSantis and co-authors

Global long-term stability of individual dietary specialization in herbivorous mammals

Larisa R. G. DeSantis^{a,b,*}, Melissa I. Pardi^b, Andrew Du^c, Michael A. Greshko^a, Lindsey T. Yann^{b,d}, Richard C. Hulbert, Jr.^e, Julien Louys^f

^aDepartment of Biological Sciences, Vanderbilt University, 1210 BSB/MRBIII 465 21st Avenue S., Nashville, TN 37232

^bDepartment of Earth and Environmental Sciences, Vanderbilt University, Science & Engineering Building, 5726 Stevenson Center; 7th floor, Nashville, TN 37240 Nashville, TN, 37235.

^cDepartment of Anthropology and Geography, Colorado State University, 1787 Campus Delivery, Fort Collins, CO 80523.

^dWaco Mammoth National Monument, 6220 Steinbeck Band Drive, Waco, TX, 76708

^eFlorida Museum of Natural History, University of Florida, Gainesville, FL, 32611.

^fAustralian Research Centre for Human Evolution, Environmental Futures Research Institute, Griffith University, Brisbane, Australia.

*Larisa R.G. DeSantis

Email: larisa.desantis@vanderbilt.edu

Classification

Biological Sciences: Ecology; Physical Sciences: Earth, Atmospheric, and Planetary Sciences

Keywords

carbon isotopes, diet, generalist, serial samples, specialist

Author Contributions

L.R.G.D. and M.A.G. conceptualized the project, M.I.P., M.A.G., and L.T.Y. compiled published data with assistance from L.R.G.D., L.R.G.D. and L.T.Y. generated new data, M.I.P. and A.D. performed statistical analysis, R.C.H. provided access to and contextual information pertaining to newly sampled specimens, L.R.G.D., M.I.P., and J.L. wrote the manuscript with contributions from A.D., M.I.P. made the figures, all authors contributed edits and intellectual contributions.

This PDF file includes:

Main Text

Figures 1 to 4

Tables 1 to 2

Abstract

[revised manuscript text omitted]

new samples of mixed-feeders (i.e., eating both grass and browse; 20 individuals, 121 samples). Enamel $\delta^{13}\text{C}$ values provide valuable insights into herbivore ecology, including a record of the proportions of C_3 and C_4 vegetation consumed, indicative of trees/shrubs and warm-season grasses, respectively—when occurring at lower latitude sites ($\sim 37^\circ$ and below; (23–25)). Because teeth grow incrementally, serial samples of enamel collected perpendicular to a tooth's growth axis can record dietary variability over the course of the tooth's growth (which can range from a few months to over two years in high-crowned teeth; (26, 27)). As a result, our new dataset allows us to answer the following questions: (1) are herbivorous generalists (i.e., mixed-feeders) composed of individual generalists (defined as having high isotopic variation) or individual specialists (defined as having low isotopic variation) (Fig. 1); and, (2) how do isotope values of individuals vary within a species' overall dietary strategy (i.e., browser, mixed-feeder, grazer)? These answers provide a long-term view of herbivore-vegetation interactions critical for understanding intra- and inter-specific competition and their ecological and evolutionary consequences, shifts in plant species composition, and species distributions through time (23, 28). Understanding how herbivores choose and consume vegetation at the individual, population, and species levels is also fundamental for effective environmental conservation and management (29), especially during the Anthropocene (30, 31).

Results

Review of the literature and stable isotope sampling of mixed-feeders resulted in 4134 serial samples from 318 individuals (Supplemental Dataset). Descriptive statistics of $\delta^{13}\text{C}_{\text{enamel}}$ values are noted for these samples in Tables 1-2, Supplemental Tables 1-9, and summarized in Figs. 1-4 and Figs. S1-S2S3.

Mammalian herbivore $\delta^{13}\text{C}_{\text{enamel}}$ values spanned 23‰ (ranging from -20.3‰ to 2.7‰) across all individuals, which includes specimens from grasslands to rainforests across the globe (e.g., Afghanistan, China, Ethiopia, Ireland, Panama, South Africa, USA; see Supplemental Dataset for a full list of the countries included). The $\delta^{13}\text{C}_{\text{enamel}}$ range within a given species can be upwards of 8‰ in grazers (Fig. 4-3), or approximately 35% of the total range represented in this study (which includes grasslands to rainforests across the globe, e.g., Afghanistan, Bulgaria, Chad, China, Ethiopia, Ireland, Jordan, Mexico, Pakistan, Panama, Scotland, South Africa, Ukraine, USA). In contrast, the range within each individual specimen is low (typically less than 3‰; Fig. 3). For example, the total species range of the mixed-feeder *Cormohipparion emsliei* from the early Pliocene (Florida, USA; (32, 33)) is 13.4‰, while individuals have an average $\delta^{13}\text{C}_{\text{enamel}}$ range of 1.1‰ (standard deviation, SD 0.5‰, $n = 9$; Fig. 2), and all individuals vary by $\leq 2\%$. An additional mixed feeder targeted for sample collection (*Hemiauchenia macrocephala*) also demonstrates a broad $\delta^{13}\text{C}_{\text{enamel}}$ range of 13.2‰ from two sites in Florida (Inglis 1A and Leisey 1A; (34)), with an average individual $\delta^{13}\text{C}_{\text{enamel}}$ range of 1.3‰ (SD of 0.5‰, $n = 8$), and all individuals exhibit $\leq 2.4\%$ variability (Fig. 1). All individual generalist herbivorous herbivores (i.e., mixed-feeders) have $\delta^{13}\text{C}_{\text{enamel}}$ ranges $\leq 4.0\%$ with average individual $\delta^{13}\text{C}_{\text{enamel}}$ ranges of $< 2\%$.

Nearly all herbivorous mammalian individuals are specialized in their diets (Fig. 3). Specifically, the majority of individuals sampled with at least three serial samples (89%, 283 of 318 individuals) or five or more serial samples (87.9%, 246 of 280 individuals) exhibit a $\delta^{13}\text{C}_{\text{enamel}}$ range $\leq 3\%$ (Fig. 3). As body size is known to influence diet, including an animal's ability to eat lower quality and foliage like grass (35), relationships between individual breadth in $\delta^{13}\text{C}_{\text{enamel}}$ and body size were examined. Specifically, we tested for significant differences in the total range of $\delta^{13}\text{C}_{\text{enamel}}$ values of small (mass < 100 kg), medium ($100 \text{ kg} \leq \text{mass} < 350$ kg), and large (mass ≥ 350 kg) species, as well as the average range exhibited by individuals across these size categories. We also reran our analyses on only specimens found below 37° latitude, where the isotopic distinction between grasses (primarily C_4) and trees/shrubs (primarily C_3) is clearest (23–

154 25). There are no significant relationships between average individual ranges or of average
taxon $\delta^{13}\text{C}_{\text{enamel}}$ ranges across body size categories, either globally or at low latitudes (Table S6),
contrary to expectations (36). However, when data were corrected with a sliding window (due to
the possibility that more serial samples could produce larger $\delta^{13}\text{C}_{\text{enamel}}$ ranges; see Materials and
Methods), the average individual range of large taxa is significantly greater than medium-sized
taxa ($p < 0.05$), but not small taxa ($p > 0.42$; Table S6).

The Isotopic Individuality Index (III) for a given species, as defined here, is the ratio of the
average individual $\delta^{13}\text{C}_{\text{enamel}}$ range to the total $\delta^{13}\text{C}_{\text{enamel}}$ range of a species. This proportion is low
(near 0) when individuals are specialized within isotopically generalized species; the III
approaches 1 when an individuals' isotopic breadth approaches the overall species breadth (Fig.
4). The average IIIs calculated for mixed-feeding species (0.18 ± 0.11 , $n=4$), browsers (0.35 ± 0.17 ,
$n = 7$), and grazers (0.39 ± 0.17 , $n=10$) are all closer to 0 than 1, with only grazers having
significantly higher III values than mixed-feeders when including the global dataset ($p < 0.05$) (Fig.
4; Table 1, S7). Below 37° latitude, the III of grazers is significantly higher than both mixed-
168 feeders ($p < 0.05$) and browsers ($p < 0.05$). It should be noted that sampling standardization (*i.e.*,
the sliding window analysis) of the global dataset results in significantly lower III values in mixed-
170 feeders as compared to browsers, $p < 0.01$ (Tables S8, S9). Variance partitioning analyses (where
the proportion of $\delta^{13}\text{C}_{\text{enamel}}$ variance is partitioned across three nested scales: between species,
between individuals, and within individuals)(37) similarly found that, across dietary groups,
individuals within species tend to be specialized (Table 2). Lower variance is consistently found
within individuals as opposed to *between-among* individuals in a species, or across species:
depending on the specific analysis used, the proportion of variance ranges from 0.34-0.64
between species, 0.32-0.61 between individuals within a species, and 0.05-0.06 between serial
samples within individuals (Table 2).

Across the global data set, the average individual $\delta^{13}\text{C}_{\text{enamel}}$ range is highest among
grazers, significantly higher than browsers (Table 1). Below 37° latitude, the average individual
$\delta^{13}\text{C}_{\text{enamel}}$ range remains highest among grazers and is significantly higher than both browsers
and mixed-feeders (Table 1). All dietary categories have mean $\delta^{13}\text{C}_{\text{enamel}}$ ranges of $< 3\text{‰}$, grazers
yielded a maximum individual range of 8‰ , while the individual $\delta^{13}\text{C}_{\text{enamel}}$ ranges of browsers and
mixed-feeders in the dataset never exceeded 3.3‰ and 3.9‰ , respectively (Fig. 3 and 4). When
we standardize the number of analyzed serial samples per tooth using a five-sample sliding
window approach (see Materials and Methods), we find similar results to the raw data analysis
(Fig. S1, Table S5). Most individual ranges (92-94%) are $\leq 2\text{‰}$ when standardized, and nearly all
186 (97-100%) are $\leq 3\text{‰}$ across dietary groups (*i.e.*, global average; Fig. S1, Table S5).

Discussion

Effects of dietary behavior on individual specialization

Herbivores that are classified as grazers vary their diet the most individually, more so than
herbivores classified as browsers and mixed-feeders; Some of the largest individual \$\delta^{13}\text{C}_{\text{enamel}}\$
ranges are exhibited by grazers (e.g., bison, horses, mammoths, and wombats can exhibit
individual \$\delta^{13}\text{C}_{\text{enamel}}\$ ranges of 6.7‰, 6.7‰, 7.6‰, and 8.0‰, respectively; Fig. S3); however,
their mean is still fairly constrained ($\bar{x} = 1.8\text{‰} \pm 1.5$ SD). As grazing morphologies permit, but do
not exclusively prescribe, grass consumption (38), dietary variability of individuals is likely to be
broader when a given taxon is capable of eating grass and/or as well as a mixed-feeding diet.
Browsing species are morphologically and nutritionally constrained to diets that exclude grasses
(39), and exhibit the lowest $\delta^{13}\text{C}_{\text{enamel}}$ ranges (Fig. 3, Table 1), which could explain the broader
 \$\delta^{13}\text{C}_{\text{enamel}}\$ ranges of grazers as compared to browsers (~1.6 to 2.4 times that of browsers; Table
1). Some of the largest individual \$\delta^{13}\text{C}_{\text{enamel}}\$ ranges are exhibited by grazers (e.g., bison, horses,
mammoths, and wombats can exhibit individual \$\delta^{13}\text{C}_{\text{enamel}}\$ ranges of 6.7‰, 6.7‰, 7.6‰, and

~~8.0‰, respectively). While many large herbivores are grazers, and body size of the individual~~
~~also impacts the plant types, parts, and volume that can be ingested (36, 39), a range of body~~
~~sizes are represented in each dietary category. Further, our results suggest that body size,~~
~~alone, is not significantly influencing individual specialization. It should be noted that while small~~
~~herbivores were analyzed (<100 kg), the smallest mammals (<10kg; e.g., most rodents) are not~~
~~included in this analysis due to the size required for serial sampling of individual teeth. While~~
~~more work is needed on small mammals, modern mark re-capture studies suggests that small~~
~~mammals can also be individually specialized (40).~~

212 The fundamental differences between grass and browse and how these
214 resources are exploited by herbivores can affect $\delta^{13}\text{C}_{\text{enamel}}$ ranges
among grazer and browser may also account for greater individual $\delta^{13}\text{C}$ ranges among grazer
species. Specifically, mammalian herbivores broadly record $\delta^{13}\text{C}_{\text{enamel}}$ variability of plants on the
landscape; depending on the size of their home range, this could introduce spatial variability in
$\delta^{13}\text{C}_{\text{enamel}}$ that may be expected to increase with body size (and inferred home range size). While
many large herbivores are classified as grazers, and body size of the individual also impacts the
plant types, parts, and volume that can be ingested (36, 39), a range of body sizes are
represented in each dietary category. Further, our results suggest that body size, alone, is not
significantly influencing individual specialization. It should be noted that while small herbivores
were analyzed (<100 kg), the smallest mammals (<10kg; e.g., most rodents) are not included in
this analysis due to the size required for serial sampling of individual teeth. While more work is
needed on small mammals, modern mark re-capture studies suggests that small mammals can
also be individually specialized (40).

Fundamental physiological differences between grasses and brose, and their distribution
across the landscape, may influence spatial heterogeneity and variation in the $\delta^{13}\text{C}_{\text{enamel}}$ values
recorded in herbivore enamel. G-Grasses typically exhibit fewer secondary plant chemicals such
as tannins and alkaloids and that can change more seasonally than leaves and buds (17),
contributing few tradeoffs for their consumption if herbivores can compensate for their typically
lower nutritional content and abrasive nature (i.e., presence of phytoliths which can wear down
teeth (41)). C₄ grasses that are more arid adapted and utilize the PEP-CK sub-pathway can also
have more negative values than grasses that use the classic NADP sub-pathway, though
differences are small (~1‰) (42). Further, from an ungulate perspective, the architectural
arrangement of grass means that it cannot be as easily differentiated via dietary selection as can
browse—grasses are composed of leaves, sheaths and fruit that differ only at the very fine scale,
they have a low growth form, and grow in continuous dispersion on the landscape (17), making it
less likely that individual grazers (typically with broad muzzles) can more finely select grass
forage. In contrast, browse consists of a heterogeneous mix of leaves, buds, and woody stems
that are irregularly distributed over the topology of the plant, and are discretely dispersed (17).
Thus, therefore, active food selection by of browse resources browsers and mixed-feeders can
lead to narrower dietary niches in browsing and mixed-feeding taxa these taxa (39) but is unlikely
to be the source of higher $\delta^{13}\text{C}_{\text{enamel}}$ variation in species consuming primarily grasses. This is
somewhat reflected in our data by the higher individuality indices (i.e., the $\delta^{13}\text{C}_{\text{enamel}}$ $\delta^{13}\text{C}$ ranges
of individuals are closer to the total $\delta^{13}\text{C}$ ranges of a given species) across of taxa classified as
grazers, though browsers have similar III values to grazers as compared to other dietary groups
likely stems from their ability to eat diverse food types in addition to grasses (with few
consequences).

Mixed-feeders, despite eating the broadest mix of food types, have the lowest III indices
(significantly less than grazers; Table 1). While we would expect lower III indices for mixed-
feeders if individuals variation is comparable to y as much as grazers or browsers, due to the
overall larger breadth represented by themixed-feeding species. However—nevertheless,
individual specialization of mixed-feeders is far below that represented by the species and does

not indicate examples of switching between the consumption of primarily grass at one interval to
the consumption of primarily browse as another. Lastly, while III indices of fossil taxa could be
lower than extant taxa due to time averaging that contributes to species breadth calculations (i.e.,
the denominator of the III values), the vast majority of all specimens included (~96%) are from
fossil specimens and are relatively comparable to one another (Supplemental Dataset SD1).

While time averaging could cause $\delta^{13}\text{C}$ ranges of extinct species to appear broader than
they may have been during a particular moment in time, it is compelling that the vast majority of
mammals exhibit narrow isotopic variability for comparable durations (months/seasons) of an
individual's life (Fig. 3, Table 1). We can't discount the possibility that herbivores change their
diet later in life or over the course of years to decades; however, serial sampling of multiple teeth
per individual (i.e., spanning years, though not decades) reveals the absence of significant dietary
variability in the majority of specimens (though this has only been documented in a handful of
studies (43-45)). Further, although stable isotopes in teeth are known to be dampened compared
to the isotopic composition of diet or water consumed (26, 4246, 4347), a given mammal tooth
has the potential to reflect the full isotopic range of a given species (e.g., have $\delta^{13}\text{C}_{\text{enamel}}$ values
that range from -10.9‰ to $0-1.4\text{‰}$). For example, bison, horses, mammoths, and wombats (all
grazers) can exhibit individual \$\delta^{13}\text{C}_{\text{enamel}}\$ ranges of 6.7‰, 6.7‰, 7.6‰, and 8.0‰, respectively
(Fig. S3) demonstrating that even when isotopes are dampened high-variability is rare but
possible. That being said, serial samples represent an averaged diet over the period of time
sampled and do not represent the full isotopic range of plants consumed by the individual during
this period of time, especially if mixed feeders are consistently consuming the same proportions
of mixed vegetation. While the total \$\delta^{13}\text{C}_{\text{enamel}}\$ range from an individual tooth is only a minimum
estimate of total plant isotopic variability (due to the signal attenuation of enamel as compared to
plants consumed by the herbivore) it does provide important insight into dietary variability of
individuals (and is a useful tool for comparing taxa with different dietary preferences as
dampening is likely to effect herbivorous mammals to a similar degree. individual teeth from *C.*
emsliei (32, 33) have a total \$\delta^{13}\text{C}_{\text{enamel}}\$ range of 13.4‰, in stark contrast to individuals which
average 1.1‰ (and all have \$\leq 2\text{‰}\$ variability in \$\delta^{13}\text{C}\$ values). While the isotopic range of plant
values may be far greater than 13.4‰, there is no reason why individual specimens of *C. emsliei*
could not vary by as much as 13.4‰ (i.e., we are comparing \$\delta^{13}\text{C}_{\text{enamel}}\$ variability of a given
individual to the \$\delta^{13}\text{C}_{\text{enamel}}\$ variability of a species). Other mixed-feeders with broad isotopic
ranges (e.g., *Hemiauchenia macrocephala*, *Platygonus vetus*, and *Diprotodon optatum* (34,
4448)) individually exhibit only a fraction of the isotopic variability of a given taxon (Fig. 4). Most
notably, very few individuals (1.2%) exhibit a broad isotopic range (i.e., individual $\delta^{13}\text{C}_{\text{enamel}}$ range
$> 7\text{‰}$), demonstrating both the possibility of high $\delta^{13}\text{C}$ variability in an individual, but also the
rarity of such occurrences.

Dietary specialists across space and time

[revised manuscript text omitted]

Of the new data collected for this publication, mixed-feeding taxa were targeted for serial sample analysis as they were rare in published literature and are of particular relevance to the research questions. All newly sampled specimens are noted in Table S1-S3, Supplementary Dataset SD1, and are housed in publicly accessible collections in the Florida Museum of Natural History (Gainesville, FL, USA). Approximately 1-2 mgs of enamel powder were drilled perpendicular to the growth axis of the tooth, using carbide drill bits and a variable speed rotary style tool. The spacing of the samples varied and is indicated in Supplementary Dataset SD1 and Table S2. Enamel powder was treated with 30% hydrogen peroxide to remove organic material, rinsed, reacted with 0.1 N acetic acid to remove diagenetic carbonates (6367), and rinsed again after 18 hours (per (34)). The treated and dried enamel powder was analyzed using a VG Prism stable isotope ratio mass spectrometer with an in-line ISOCARB automatic sampler in the Department of Geological Sciences at the University of Florida. The analytical precision is $\pm 0.1\text{‰}$, based on replicate analyses of samples and standards (NBS-19). Stable isotope data are reported in conventional delta (δ) notation for carbon ($\delta^{13}\text{C}$) and oxygen ($\delta^{18}\text{O}$), where $\delta^{13}\text{C}$ (parts per mil, ‰) = $((R_{\text{sample}}/R_{\text{standard}}) - 1) * 1000$, and $R = {}^{13}\text{C}/{}^{12}\text{C}$; $\delta^{18}\text{O}$ (parts per mil, ‰) = $((R_{\text{sample}}/R_{\text{standard}}) - 1) * 1000$, and $R = {}^{18}\text{O}/{}^{16}\text{O}$; and the standard is VPDB (Pee Dee Belemnite, Vienna Convention) (6468). All stable carbon isotopes are from the carbonate portion of tooth enamel hydroxylapatite.

Dietary classification—Each taxon was categorized as a browser, grazer, or mixed-feeder according to literature consensus. For extant species, this was determined by the observed diet of wild caught specimens (note, only wild caught specimens were included in the database; hence, no domesticated species were included from modern or archeological sites). For extinct species, these determinations were made based on studies of analogous morphology with living species. Taxon names, diets, and the references used to justify the assignments for extinct taxa are given in Table S10.

Statistical analyses—The minimum, maximum, range, mean, and standard deviation of the measured $\delta^{13}\text{C}_{\text{enamel}}$ serial samples were calculated for each individual included in the database (Supplementary Dataset S1). When multiple publications reported data from the same specimen, values from the earliest publication were used and referenced. When serial samples were from multiple teeth from the same individual, summary statistics were calculated from the combined serial samples of the individual. Individuals were analyzed if they had at least three serial samples for a given tooth.

To explore whether diet type influences individual $\delta^{13}\text{C}_{\text{enamel}}$ breadth, individual range was
categorized into 1‰ bins (i.e.; $0 < x \leq 1\text{‰}$, $1 < x \leq 2\text{‰}$, etc.), and the proportion of individuals
occupying each dietary type was calculated for each bin. To assess the influence of body size on
individual breadth in $\delta^{13}\text{C}_{\text{enamel}}$, we tested for significant differences in the range of $\delta^{13}\text{C}_{\text{enamel}}$
species of small (mass <100 kg), medium (100 kg \leq mass <350 kg), and large (mass \geq 350 kg)
body size as well as the average range of individuals across these size categories. These
categories are based on the commonly used and well-established body size categories of
herbivorous mammals, based in part on dietary differences and also the prey sizes readily
consumed by African predators (69-71). Body size estimates were made from the published
literature, or new estimates were made based on allometries (Table S11). The potential influence
of body size was assessed for the global dataset, the global dataset corrected for serial sampling
bias, the dataset restricted to low latitudes (<37°), and the low latitude dataset corrected for serial
sampling bias. Significant differences between average individual ranges according to dietary
type was tested using Dunn's test of non-parametric pairwise multiple comparisons.

Grazers tend to have higher-crowned teeth than browsers and mixed-feeders, and
therefore tend to have more serial samples per tooth. To account for the possibility that more
serial samples might produce a larger range of $\delta^{13}\text{C}_{\text{enamel}}$ values, we performed a moving window
analysis for each tooth, where we iteratively calculated the $\delta^{13}\text{C}_{\text{enamel}}$ range for five consecutive
samples, as denoted by each sample's distance from root along the tooth's growth axis. As an
example, for a given tooth, we subset out samples 1-5, where one represents the sample closest
to the root, and calculated the range of $\delta^{13}\text{C}_{\text{enamel}}$ values. We then subset out samples 2-6, 3-7,
and so on until the sample furthest from the root was included in a window. If multiple teeth were
sampled from an individual, this procedure was applied separately to each tooth.

An Isotopic Individuality Index (III) was calculated for each taxon, as follows: Average
individual range of $\delta^{13}\text{C}_{\text{enamel}}$ (for a given taxon)/ Total $\delta^{13}\text{C}_{\text{enamel}}$ range of the taxon. Average IIIs
for dietary categories were calculated using only species-level data, unless it could be reasonably
inferred that a genus was represented in the database by only one species (e.g., *Camelops* sp. in
North America). Average IIIs were calculated at two scales: 1) at the global scale, and 2)
restricted to individuals collected below 37° latitude (where C₄ resources are primarily warm-
season grasses and C₃ resources are primarily trees and shrubs, in contrast to C₃ ecosystems
above 37° latitude where grass and browse cannot be discerned from isotopes alone (24, 25)).
While we recognize that the latitudinal gradient in C₃ and C₄ grasses is likely a recent
phenomenon, since ~5-7 Ma (23, 6572), there is no evidence from our analysis that any taxon is
switching from C₃ to C₄ grass, seasonally or during the period of time during which their teeth
mineralize (i.e., no teeth vary by ~14‰, the mean difference between C₃ and C₄ grasses; (23)).
Thus, the analysis of $\delta^{13}\text{C}_{\text{enamel}}$ values of individuals collected below 37° latitude is a reasonable
proxy for inferring individual dietary variability.

We conducted variance partitioning analyses to quantify the proportion of
$\delta^{13}\text{C}_{\text{enamel}}$ variance that can be found between species, between individuals, and within
individuals. We only analyzed those teeth with at least five serial samples. Individuals identified to
the genus level were lumped with congener individuals identified to the species level only when it
could be reasonably inferred based on location and geological time-period that they belonged to
the same species (e.g.i.e., *Mammuthus* sp. lumped with *Mammuthus columbi*). Variance
partitioning was done in two ways using R 4.0.2 (6673). (1) We fit via restricted maximum
likelihood an intercept-only, nested ANOVA with $\delta^{13}\text{C}_{\text{enamel}}$ as the dependent variable and
individuals nested within species as random effects. This was done using the lme() function in the
"nlme" R package (6774), and the varcomp() function in the "ape" package (6875) was used to
extract the variance components. (2) We also partitioned variance (sum of squares) in
 $\delta^{13}\text{C}_{\text{enamel}}$ using two-level, nested ANOVAs for unbalanced data following Ref. (37) (their Box
10.6, pp. 294-298).

Acknowledgments

We are grateful to the numerous museums and associated staff who have allowed researchers to serial sample specimens included in this study, including the Florida Museum of Natural History for allowing the collection of new data (B. MacFadden and J. Bloch) and J. Curtis (University of Florida) for analytical assistance. We are thankful to the faculty, staff, and students at Vanderbilt University, and numerous researchers for conversations regarding these data (including participants and organizers of the 2nd Lembersky Conference in Human Evolutionary Studies, Advances in Paleoecology Conference, 2018, R. Scott and A. Barr). The work was funded by the National Science Foundation (EAR 1725154) and Vanderbilt University.

**References**

- 1. D. S. Gardner, S. E. Ozanne, K. D. Sinclair, Effect of the early-life nutritional environment on fecundity and fertility of mammals. *Philos. Trans. R. Soc. B Biol. Sci.* **364**, 3419–3427 (2009).
- 2. N. Owen-Smith, J. M. Fryxell, E. H. Merrill, Foraging theory upscaled: The behavioural ecology of herbivore movement. *Philos. Trans. R. Soc. B Biol. Sci.* **365**, 2267–2278 (2010).
- 3. J. G. C. Hopcraft, *et al.*, “Why are wildebeest the most abundant herbivore in the Serengeti ecosystem?” in *Serengeti IV: Sustaining Biodiversity in a Coupled Human-Natural System*, (2015), pp. 125–174.
- 4. A. M. Wilson, *et al.*, Biomechanics of predator-prey arms race in lion, zebra, cheetah and impala. *Nature* **554**, 183–188 (2018).
- 5. P. Roberts, B. A. Stewart, Defining the ‘generalist specialist’ niche for Pleistocene *Homo sapiens*. *Nat. Hum. Behav.* **2**, 542–550 (2018).
- 6. L. Van Valen, Morphological Variation and Width of Ecological Niche. *Am. Nat.* **99**, 377–390 (1965).
- 7. D. I. Bolnick, *et al.*, The ecology of individuals: Incidence and implications of individual specialization. *Am. Nat.* **161**, 1–28 (2003).
- 8. D. I. Bolnick, R. Svanbäck, M. S. Araújo, L. Persson, Comparative support for the niche variation hypothesis that more generalized populations also are more heterogeneous. *Proc. Natl. Acad. Sci. U. S. A.* **104**, 10075–10079 (2007).
- 9. P. Matich, M. R. Heithaus, C. A. Layman, Contrasting patterns of individual specialization and trophic coupling in two marine apex predators. *J. Anim. Ecol.* **80**, 294–305 (2011).
- 10. H. B. V. Zanden, K. A. Bjorndal, K. J. Reich, A. B. Bolten, Individual specialists in a generalist population: Results from a long-term stable isotope series. *Biol. Lett.* **6**, 711–714 (2010).
- 11. S. D. Newsome, *et al.*, Using stable isotopes to investigate individual diet specialization in California sea otters (*Enhydra lutris nereis*). *Ecology* **90**, 961–974 (2009).
- 12. K. J. Woo, K. H. Elliott, M. Davidson, A. J. Gaston, G. K. Davoren, Individual specialization in diet by
a generalist marine predator reflects specialization in foraging behaviour. *J. Anim. Ecol.* **77**, 1082–1091 (2008).
- 13. E. J. M. Urton, K. A. Hobson, Intrapopulation variation in gray wolf isotope ($\delta^{15}\text{N}$ and $\delta^{13}\text{C}$) profiles: Implications for the ecology of individuals. *Oecologia* **145**, 317–326 (2005).
- 14. M. E. Hanley, Seedling herbivory, community composition and plant life history traits. *Perspect. Plant Ecol. Evol. Syst.* **1**, 191–205 (1998).
- 15. A. M. Felton, *et al.*, The nutritional balancing act of a large herbivore: An Experiment with Captive Moose (*Alces alces* L). *PLoS One* **11**, e0150870 (2016).
- 16. C. McArthur, C. T. Robbins, A. E. Hagerman, T. A. Hanley, Diet selection by a ruminant generalist browser in relation to plant chemistry. *Can. J. Zool.* **71**, 2236–2243 (1993).
- 17. L. Shipley, Grazers and browsers: how digestive morphology affects diet selection. *Grazing Behav. Livest. Wildl.*, 20–27 (1999).
18. J. M. Emlen, The Role of Time and Energy in Food Preference. *Am. Nat.* **100**, 611–617 (1966).
- 19. R. H. MacArthur, E. R. Pianka, On Optimal Use of a Patchy Environment. *Am. Nat.* **100**, 603–609 (1966).
- 20. G. H. Pyke, H. R. Pulliam, E. L. Charnov, Optimal Foraging: A Selective Review of Theory and Tests. *Q. Rev. Biol.* **52**, 137–154 (1977).

- 21. G. H. Pyke, Optimal foraging theory: a critical review. *Annu. Rev. Ecol. Syst. Vol. 15*, 523–575 (1984).
- 22. S. L. Wilson, G. I. H. Kerley, Bite diameter selection by thicket browsers: The effect of body size and plant morphology on forage intake and quality. *For. Ecol. Manage.* **181**, 51–65 (2003).
- 23. T. E. Cerling, *et al.*, Global vegetation change through the Miocene/Pliocene boundary. *Nature* **389**, 153–158 (1997).
- 24. J. A. Teeri, L. G. Stowe, Climatic patterns and the distribution of C4 grasses in North America. *Oecologia* **23**, 1–12 (1976).
- 25. L. G. Stowe, J. A. Teeri, The Geographic Distribution of C 4 Species of the Dicotyledonae in Relation to Climate. *Am. Nat.* **112**, 609–623 (1978).
- 26. B. H. Passey, T. E. Cerling, Tooth enamel mineralization in ungulates: Implications for recovering a primary isotopic time-series. *Geochim. Cosmochim. Acta* **66**, 3225–3234 (2002).
- 27. K. A. Hoppe, R. Amundson, M. Vavra, M. P. McClaran, D. L. Anderson, Isotopic analysis of tooth enamel carbonate from modern North American feral horses: Implications for paleoenvironmental reconstructions. *Palaeogeogr. Palaeoclimatol. Palaeoecol.* **203**, 299–311 (2004).
- 28. D. I. Bolnick, *et al.*, Why intraspecific trait variation matters in community ecology. *Trends Ecol. Evol.* **26**, 183–192 (2011).
- 29. P. J. Weisberg, H. Bugmann, Forest dynamics and ungulate herbivory: From leaf to landscape. *For. Ecol. Manage.* **181**, 1–12 (2003).
- 30. S. L. Lewis, M. A. Maslin, Defining the Anthropocene. *Nature* **519**, 171–180 (2015).
- 31. C. N. Waters, *et al.*, The Anthropocene is functionally and stratigraphically distinct from the Holocene. *Science (80-.)*. **351**, aad2622 (2016).
- 32. R. C. Hulbert, A new Cormohipparion (Mammalia, Equidae) from the pliocene (latest Hemphillian and Blancan) of Florida. *J. Vertebr. Paleontol.* **7**, 451–468 (1988).
- 33. S. D. Webb, R. C. Hulbert Jr., G. S. Morgan, H. F. Evans, Terrestrial mammals of the Palmetto Fauna (early Pliocene, latest Hemphillian) from the central Florida phosphate district. *Nat. Hist. Museum Los Angeles Cty. Sci. Ser.* **41**, 293–312 (2008).
- 34. L. R. G. DeSantis, R. S. Feranec, B. J. MacFadden, Effects of global warming on ancient mammalian communities and their environments. *PLoS One* **4**, e5750 (2009).
- 35. A. W. Illius, I. J. Gordon, Modelling the nutritional ecology of ungulate herbivores: evolution of body size and competitive interactions. *Oecologia* **89**, 428–434 (1992).
- 36. J. Codron, *et al.*, Stable isotope series from elephant ivory reveal lifetime histories of a true dietary generalist. *Proc. R. Soc. B Biol. Sci.* **279**, 2433–2441 (2012).
- 37. R. R. Sokal, F. J. Rohlf, *Biometry: the principles and practice of statistics in biological research. 3rd Edition.* (W.H. Freeman, 1994).
- 38. R. S. Feranec, Stable isotopes, hypsodonty, and the paleodiet of Hemiauchenia (Mammalia: Camelidae): A morphological specialization creating ecological generalization. *Paleobiology* **29**, 230–242 (2003).
- 39. K. R. Searle, L. A. Shipley, “The Comparative Feeding Behaviour of Large Browsing and Grazing Herbivores” in *The Ecology of Browsing and Grazing*, (Springer, 2008), pp. 117–148.
- 40. J. D. Noble, *et al.*, Foraging strategies of individual silky pocket mice over a boom–bust cycle in a stochastic dryland ecosystem. *Oecologia* **190**, 569–578 (2019).

- 41. C. A. E. Strömberg, V. S. Di Stilio, Z. Song, Functions of phytoliths in vascular plants: an evolutionary perspective. *Funct. Ecol.* **30**, 1286–1297 (2016).
- 42. T. E. Cerling, J. M. Harris, Carbon isotope fractionation between diet and bioapatite in ungulate mammals and implications for ecological and paleoecological studies. *Oecologia* **120**, 347-363 (1999).
- 43. M. Balasse, S. H. Ambrose, A. B. Smith, T. D. Price, The seasonal mobility model for prehistoric herders in the south-western Cape of South Africa assessed by isotopic analysis of sheep tooth enamel. *J. Archaeol. Sci.* **29**, 917–932 (2002).
- 44. A. Zazzo, A. Mariotti, C. Lécuyer, E. Heintz, Intra-tooth isotope variations in late Miocene bovid enamel from Afghanistan: paleobiological, taphonomic, and climatic implications. *Palaeogeogr. Palaeoclimatol. Palaeoecol.* **186**, 145-161 (2002).
- 45. J. A. van Dam, G. J. Reichart, Oxygen and carbon isotope signatures in late Neogene horse teeth from Spain and application as temperature and seasonality proxies. *Palaeogeogr. Palaeoclimatol. Palaeoecol.* **274**, 64-81 (2009).
- 4246. B. H. Passey, T. E. Cerling, Response to the comment by M. J. Kohn on “Tooth enamel mineralization in ungulates: Implications for recovering a primary isotopic time-series,” by B. H. Passey and T. E. Cerling (2002). *Geochim. Cosmochim. Acta* **68**, 407–409 (2004).
- 4347. M. J. Kohn, Comment: Tooth enamel mineralization in ungulates: Implications for recovering a primary isotopic time-series, by B. H. Passey and T. E. Cerling (2002). *Geochim. Cosmochim. Acta* **68**, 403–405 (2004).
- 4448. L. R. G. Desantis, J. H. Field, S. Wroe, J. R. Dodson, Dietary responses of Sahul (Pleistocene Australia-New Guinea) megafauna to climate and environmental change. *Paleobiology* **43**, 181–195 (2017).
- 4549. D. Réale, S. M. Reader, D. Sol, P. T. McDougall, N. J. Dingemanse, Integrating animal temperament within ecology and evolution. *Biol. Rev.* **82**, 291–318 (2007).
- 4650. M. Wolf, F. J. Weissing, Animal personalities: Consequences for ecology and evolution. *Trends Ecol. Evol.* **27**, 452–461 (2012).
- 4751. C. Darwin, *On the Origin of Species, 1859* (1859) <https://doi.org/10.4324/9780203509104>.
- 4852. P. Michelena, R. Jeanson, J. L. Deneubourg, A. M. Sibbald, Personality and collective decision-making in foraging herbivores. *Proc. R. Soc. B Biol. Sci.* **277**, 1093–1099 (2010).
- 4953. J. W. Jolles, L. Ostojčić, N. S. Clayton, Dominance, pair bonds and boldness determine social-foraging tactics in rooks, *Corvus frugilegus*. *Anim. Behav.* **85**, 1261–1269 (2013).
- 5054. R. H. J. M. Kurvers, B. A. Nolet, H. H. T. Prins, R. C. Ydenberg, K. Van Oers, Boldness affects foraging decisions in barnacle geese: An experimental approach. *Behav. Ecol.* **23**, 1155–1161 (2012).
- 5155. R. H. J. M. Kurvers, S. I. Van Santen De Hoog, S. E. Van Wieren, R. C. Ydenberg, H. H. T. Prins, No evidence for negative frequency-dependent feeding performance in relation to personality. *Behav. Ecol.* **23**, 51–57 (2012).
- 5256. N. P. Moran, B. B. M. Wong, R. M. Thompson, Weaving animal temperament into food webs: implications for biodiversity. *Oikos* **126**, 917–930 (2017).
- 5357. R. Svanbäck, D. I. Bolnick, Intraspecific competition drives increased resource use diversity within a natural population. *Proc. R. Soc. B Biol. Sci.* **274**, 839–844 (2007).
- 5458. M. Doebeli, Genetic variation and the persistence of predator-prey interactions in the Nicholson-Bailey model. *J. Theor. Biol.* **188**, 109–120 (1997).

- 5559. R. Bürger, A multilocus analysis of intraspecific competition and stabilizing selection on a quantitative trait. *J. Math. Biol.* **50**, 355–396 (2005).
- 5660. D. I. Bolnick, *et al.*, Ecological release from interspecific competition leads to decoupled changes in population and individual niche width in *Proceedings of the Royal Society B: Biological Sciences*, (2010), pp. 1789–1797.
- 5761. A. M. Torregrossa, A. V. Azzara, M. D. Dearing, Testing the diet-breadth trade-off hypothesis: Differential regulation of novel plant secondary compounds by a specialist and a generalist herbivore. *Oecologia* **168**, 711–718 (2012).
- 5862. D. U. Hooper, *et al.*, Effects of biodiversity on ecosystem functioning: A consensus of current knowledge. *Ecol. Monogr.* **75**, 3–35 (2005).
- 5963. D. Jablonski, Species selection: Theory and data. *Annu. Rev. Ecol. Evol. Syst.* **39**, 501–524 (2008).
- 6064. D. Jablonski, Approaches to Macroevolution: 2. Sorting of Variation, Some Overarching Issues, and General Conclusions. *Evol. Biol.* **44**, 451–475 (2017).
- 6465. P. D. Smits, Expected time-invariant effects of biological traits on mammal species duration. *Proc. Natl. Acad. Sci. U. S. A.* **112**, 13015–13020 (2015).
- 6266. W. J. Ripple, *et al.*, Collapse of the world’s largest herbivores. *Sci. Adv.* **1**, e1400103 (2015).
- 6367. P. L. Koch, N. Tuross, M. L. Fogel, The effects of sample treatment and diagenesis on the isotopic integrity of carbonate in biogenic hydroxylapatite. *J. Archaeol. Sci.* **24**, 417–429 (1997).
- 6468. T. B. Coplen, Reporting of stable hydrogen, carbon, and oxygen isotopic abundances: (Technical Report). *Pure Appl. Chem.* **66**, 273–276 (1994).
- 69. H. T. Bunn III, *Meat-eating and human evolution: studies on the diet and subsistence patterns of Plio-Pleistocene hominids in East Africa*. Doctoral dissertation, University of California, Berkeley (1982).
- 70. C. K. Brain, *The hunters or the hunted?: An introduction to African cave taphonomy*. University of Chicago Press (1983).
- 71. J. T. Faith, J. Rowan, A. Du, Early hominins evolved within non-analog ecosystems. *Proc. Natl. Acad. Sci. U. S. A.* **116**, 21478–21483 (2019).
- 6572. B. J. Macfadden, T. E. Cerling, J. M. Harris, J. Prado, Ancient latitudinal gradients of C3/C4 grasses interpreted from stable isotopes of New World Pleistocene horse (*Equus*) teeth. *Glob. Ecol. Biogeogr.* **8**, 137–149 (1999).
- 6673. R. D. C. T. 3.5.1., *A Language and Environment for Statistical Computing* (2018).
- 6774. J. Pinheiro, D. Bates, S. DebRoy, D. Sarkar, R Core Team (2014). nlme: linear and nonlinear mixed effects models. R package version 3.1–117. URL <http://cran.r-project.org/web/packages/nlme/index.html> (2014).
- 6875. E. Paradis, K. Schliep, Ape 5.0: An environment for modern phylogenetics and evolutionary analyses in R. *Bioinformatics* **35**, 526–528 (2019).

Figure 1. Stable carbon isotopic breadth of grazing, mixed-feeding, and browsing species. Hypothetical examples denote expected individual variability if individuals are specialists (left) or more generalized (middle). Empirical data (right, examples of serial samples from Supplementary Dataset) indicate that the average $\delta^{13}\text{C}_{\text{enamel}}$ range is 1.4‰ (1.1 SD, n=21; Supplementary Tables S1 and S2); grazing (green), mix-feeding (orange), browsing (blue).

**Figure 2.** Carbon stable isotope values from individual specimens of *Cormohipparion emsliei*
 from a 5 million-year-old fossil assemblage (Bone Valley, Florida, USA). Bulk and mean serial
 values (left) along with raw serial samples (right) exemplify broad dietary breadth as a species
 (range = 13.4‰), while individuals are highly specialized (all individuals sampled have $\leq 2\%$
 range of $\delta^{13}\text{C}_{\text{enamel}}$, with an average $\delta^{13}\text{C}_{\text{enamel}}$ range of 1.1‰ Supplementary Tables S2 and S3).
 **O**bligat**e** **b**rowsers (blue), **o**bligat**e** **g**razers (green), and all remaining mixed-feeders are
 indicated with other colors (orange-purple).

654

Figure 3. Proportion of individuals within serial-sample range bins per dietary group. Most serially sampled individuals (82.4–89%) regardless of dietary category, have $\delta^{13}C_{enamel}$ values that range $\leq 3\%$ across when calculated using the full range of statistical methods. Only a small proportion of individuals (3.8–6%) range in $\delta^{13}C_{enamel}$ values by more than 5‰, all of which are grazers. The right skewed pattern is present whether we use A) the entire dataset, B) individuals with 5 or more serial-samples, or when we consider C) all samples below 37° latitude, and D) individuals that have at least 5 serial samples that are also restricted to below 37° latitude.

**Figure 4.** Isotopic Individuality Index (III) per dietary category. A) The overall species dietary
 range plotted against the average individual range for a species (one-to-one line given, species
 codes are defined in Table S7), and B) the distribution of III within dietary categories indicate a
 high degree of individual specialization (low index). Values closer to one are species in which the
 individuals are each more representative of the species as a whole, while smaller values indicate
 that individuals are specialized and less representative of their overall species dietary breadth. C)
 Below 37° latitude grazers plot near the one-to-one line more often than other groups; D) the III

for grazing species is higher than mixed-feeders below 37° latitude, and browsers at these lower
latitudes are uncommon in this study.

**Table 1.** Summary statistics of dietary ranges for individuals, Isotopic Individuality Indices (III), and subsequent comparisons between groups using Kruskal–Wallis and Dunn’s tests.

Dietary Category	Mean	SD	n	Comparison	p-value
Global Dataset		Range			
Browser	1.1	0.8	48	Browser vs. Grazer	<0.001
Mixed-Feeder	1.4	0.9	33	Mixed-Feeder vs. Browser	0.0629
Grazer	1.8	1.5	231	Grazer vs. Mixed-Feeder	0.3785
Below 37° Latitude		Range			
Browser	1	0.7	19	Browser vs. Grazer	<0.00001
Mixed-Feeder	1.5	1.0	32	Mixed-Feeder vs. Browser	0.0815
Grazer	2.4	1.7	111	Grazer vs. Mixed-Feeder	<0.01
Global Dataset		III			
Browser	0.35	0.17	7	Browser vs. Grazer	0.5530
Mixed-Feeder	0.18	0.11	4	Mixed-Feeder vs. Browser	0.1207
Grazer	0.39	0.17	10	Grazer vs. Mixed-Feeder	0.0325
Below 37° Latitude		III			
Browser	0.15	0.04	2	Browser vs. Grazer	0.0475
Mixed-Feeder	0.18	0.11	4	Mixed-Feeder vs. Browser	0.9362
Grazer	0.44	0.11	6	Grazer vs. Mixed-Feeder	0.0164

SD, standard deviation; n, number of individuals (in reference to range), number of species (in reference to III). Bold p-values are significant, <0.05.

**Table 2.** Variance partitioning analyses of individuals having at least five serial samples. Data are
analyzed using both the entire dataset and samples from below 37° latitude. Treatment of
individuals that were only identified to genus (published as "sp.") were "lumped" with congener
individuals identified to the species level when it could be reasonably inferred that they belonged
to the same species. The alternative was to treat "sp." "as is", or as their own taxon. The "REML"
method uses restricted maximum likelihood fit a random effects nested ANOVA, while "SS"
partitions sum of squares using nested ANOVAs for unbalanced data. The various levels at which
variance is located are given, with the proportion of variance given for each dataset type,
treatment, and method.

Dataset	Treatment	Method	Between Species	Between Individuals	Within Individuals
Global	As Is	REML	0.29	0.67	0.04
Global	As Is	SS	0.31	0.64	0.04
Global	Lumped	REML	0.34	0.61	0.05
Global	Lumped	SS	0.57	0.38	0.05
Below 37°	As Is	REML	0.42	0.53	0.05
Below 37°	As Is	SS	0.38	0.57	0.05
Below 37°	Lumped	REML	0.43	0.51	0.06
Below 37°	Lumped	SS	0.63	0.32	0.06

688